# Variation of connectivity across exemplar sensory and associative thalamocortical loops in the mouse

Arghya Mukherjee[1,2†]*, Navdeep Bajwa[1,2†], Norman H Lam[1,2], César Porrero[3], Francisco Clasca[3], Michael M Halassa[1,2]*

[1]McGovern Institute for Brain Research, Cambridge, United States; [2]Department of Brain and Cognitive Sciences, Massachusetts Institute of Technology, Cambridge, United States; [3]Department of Anatomy and Neuroscience, School of Medicine, Autónoma de Madrid University, Madrid, Spain

**Abstract** The thalamus engages in sensation, action, and cognition, but the structure underlying these functions is poorly understood. Thalamic innervation of associative cortex targets several interneuron types, modulating dynamics and influencing plasticity. Is this structure-function relationship distinct from that of sensory thalamocortical systems? Here, we systematically compared function and structure across a sensory and an associative thalamocortical loop in the mouse. Enhancing excitability of mediodorsal thalamus, an associative structure, resulted in prefrontal activity dominated by inhibition. Equivalent enhancement of medial geniculate excitability robustly drove auditory cortical excitation. Structurally, geniculate axons innervated excitatory cortical targets in a preferential manner and with larger synaptic terminals, providing a putative explanation for functional divergence. The two thalamic circuits also had distinct input patterns, with mediodorsal thalamus receiving innervation from a diverse set of cortical areas. Altogether, our findings contribute to the emerging view of functional diversity across thalamic microcircuits and its structural basis.

*For correspondence:
mukhargh@mit.edu (AM);
mhalassa@mit.edu (MMH)

†These authors contributed equally to this work

Competing interests: The authors declare that no competing interests exist.

## Introduction

Interactions between the thalamus and cortex are essential for sensory processing (*Lien and Scanziani, 2018*; *Bartlett, 2013*), motor control (*Fresno et al., 2019*; *Guo et al., 2017*), arousal (*Chen et al., 2015*; *Schiff, 2008*) and multiple aspects of cognitive function (*Wimmer et al., 2015*; *Wolff and Vann, 2019*; *Jaramillo et al., 2019*; *Parnaudeau et al., 2018*; *Pergola et al., 2018*; *Rikhye et al., 2018a*; *Saalmann and Kastner, 2011*; *Bolkan et al., 2017*; *Scott et al., 2020*). How does this collection of forebrain nuclei contribute to the diversity of hypothesized functions? The thalamus is primarily composed of excitatory neurons that are devoid of recurrent local connections, giving way to the notion that the computations thalamic microcircuits perform will have to rely on dynamic changes in their inputs (*Halassa and Kastner, 2017*; *Jaramillo et al., 2019*). In early sensory pathways such as the geniculocortical, lateral geniculate thalamic neurons receive strong and orderly segregated inputs from subcortical sources (*Hubel and Wiesel, 1961*; *Usrey et al., 1999*), making their responses similar to their inputs (but see *Liang et al., 2018*), and suggesting a relatively limited computational role (*Lien and Scanziani, 2013*; *Priebe and Ferster, 2012*; *Reid and Alonso, 1996*; *Li et al., 2013b*; *Li et al., 2013a*). In fact, because cortical neurons in primary cortical areas show sensory responses that are qualitatively distinct but explained by a weighted sum of the thalamic inputs they receive (*Finn et al., 2007*; *Alonso et al., 2007*; *Lien and Scanziani, 2013*; *Usrey and Alitto, 2015*), the sensory thalamic microcircuits are thought to function primarily as *relays* (*Guillery and Sherman, 2002*; *Sherman, 2016*).

Despite the paucity of data on thalamic microcircuit structural organization beyond these select sensory pathways, the notion that the thalamus is largely a relay station permeates the literature on thalamocortical interactions (*Mitchell, 2015*; *Moustafa et al., 2017*; *Sherman, 2017*). For example, neurons in areas like the posterior Pulvinar nuclear complex or the Mediodorsal nucleus, which constitutes a large part of the human thalamus (*Chalfin et al., 2007*) have been hypothesized to operate as *higher order relays*, transmitting signals from one cortical area to another. Although there is some experimental support for this idea (*Guillery and Sherman, 2002*; *Sherman, 2016*; *Sherman and Guillery, 2002*), whether the higher order relay function is universal, or describes only a subset of the computations carried out in these nuclei is an open question. In fact, a number of functional studies suggest that certain responses in the pulvinar and posterior nucleus maybe explained by convergent cortical and/or subcortical inputs (*Groh et al., 2014*; *Komura et al., 2013*; *Roth et al., 2016*), indicating that at least some of these microcircuits may be performing computations on their inputs (*Bieler et al., 2018*; *Mease et al., 2017*; *Whitmire et al., 2016*). In addition, a recent study in the mouse has shown that pulvinar neurons may be preferentially innervating inhibitory neurons in primary visual cortex, contributing to surround suppression of their sensory responses (*Fang et al., 2020*). Together, these studies challenge the notion of these nuclei functioning uniformly as *higher order* relays. More generally, a number of open questions exist regarding the wiring diversity of thalamic microcircuits and the computations that they perform (*Halassa and Sherman, 2019*; *Nakajima and Halassa, 2017*).

A rapidly growing field of research into the role of the thalamus beyond sensory processing, into topics such as motor control and cognitive flexibility (*Wolff and Vann, 2019*; *Fresno et al., 2019*; *Rikhye et al., 2018a*), is highlighting the relevance of non-relay thalamocortical computations. For example, thalamocortical neurons in the ventromedial thalamus are engaged in maintaining preparatory motor activity in premotor cortex of the mouse (*Guo et al., 2017*). In primates, a speed control function in regulating cortical preparatory dynamics has been described for the ventroanterior/ventrolateral thalamus (*Wang et al., 2018*). These types of functions are much better described from a control theory perspective (*Kao et al., 2020*), where the thalamus constrains cortical dynamics rather than transmit signals as in early sensory systems (*Nakajima and Halassa, 2017*).

Similar to studies of motor control, a variety of studies focused on cognitive processing have either directly demonstrated or inferred a non-relay function of thalamic microcircuits. For example, the mediodorsal thalamus (MD), the largest thalamic input to the prefrontal cortex (PFC) has been shown to be critical for maintaining cognitive control signals underlying attention (*Schmitt et al., 2017*), adaptive decision making (*Alcaraz et al., 2018*) and working memory (*Bolkan et al., 2017*). In addition, this thalamic structure is also critical for enabling animals to flexibly switch between different rules that guide behavioral choice (*Rikhye et al., 2018a*). One study (*Rikhye et al., 2018a*) has shown that the MD thalamus contains a functional subpopulation that exerts a suppressive impact on prefrontal activity, enabling its representations to switch according to shifts in behavioral context. The anatomical circuit substrates underlying such functionality are poorly understood.

Motivated by these functional studies, we examined in mice the reciprocal connectivity of MD neurons with their targets in the prelimbic part of the PFC (PL), a hypothesized rodent analogue of primate dorsolateral PFC (*Hoover and Vertes, 2007*; *Vertes, 2004*). Given the notion that this thalamocortical loop engages in a variety of cognitive control, non-relay functions (*Ferguson and Gao, 2018*; *Schmitt and Halassa, 2017*), we systematically compared the impact of MD activation on PL activity, as well as its input/output connectivity patterns to those of a classical sensory one. We chose the medial geniculate pathway to primary auditory cortex (A1), given the ease by which this pathway can be controlled in rodents (*Hackett et al., 2011*). As such, we performed a series of functional and anatomical studies using classical and state-of-the-art tools to systematically compare those two thalamocortical loops. Sustained drive of MD neurons broadly over several hundred milliseconds, using stabilized step-function opsins (SSFOs), resulted in prelimbic activity patterns that were dominated by inhibition. This was quite distinct from an equivalent manipulation of the MGB, where A1 excitatory drive was robust. These functional differences were explained by a series of anatomical findings including unique patterns of excitatory/inhibitory cortical innervation across the two loops. More specifically, the medial geniculate body (MGB) innervated excitatory A1 neurons in higher proportions compared to the manner by which MD terminals innervated PL microcircuits. In addition, we observed a striking difference in the size of MD vs. MGB synaptic terminals selectively labeled using BDA as well as genetic labeling using mammalian green fluorescence protein [GFP]

reconstitution across synaptic partners (mGRASP *Feng et al., 2014*). In addition to these output structural differences, we also saw difference on the input side, where the major excitatory inputs to the MD are from frontal cortical areas, compared to MGB that receives excitatory drive mostly from the midbrain. Based on the origin of their driving inputs our data is consistent with the notion of the MD as a *higher order* nucleus and the MGB as *a first order* nucleus (*Mitchell, 2015*; *Sherman and Guillery, 1998*). They are also in line with the broader notion of sensory thalamus being modulated by its cortical target, while associative thalamus being driven by its cortical target (*Crick and Koch, 1998*). Overall, our study provides structural and functional explanation for why some thalamocortical loops perform a relay function, while others are more suited for integrative and cognitive control functions.

## Results

### Driving the MD has a distinct impact on PL compared to MGB's impact on A1

To compare the functional impact of MD activation on the PL with that of the MGB on A1, we combined multi-site multielectrode recordings across these two thalamocortical loops with causal manipulations (*Figure 1A,B*; see Materials and methods). We sought to uniformly enhance excitability in either the MD or MGB, akin to a step current injection. Therefore, in two groups of animals (N = 5 animals per condition), we transduced either the MD (*Figure 1A*) or MGB (*Figure 1B*) with stabilized step function opsins (SSFO, hChR2(C128S/D156A)), an optical actuator known to precisely accomplish this goal (*Yizhar et al., 2011*). Using stereotactic injections (see Materials and methods), we preferentially targeted the lateral MD and the ventral portion of the MGB, because these areas are known to primarily target the PL (*Groenewegen, 1988*) and A1 (*Winer et al., 2005*), respectively.

Across each animal, we ensured that optical stimulation resulted in comparable increase in thalamic spiking (spiking enhancement: MD = 66.18 +/- 1.89% (mean +/- SEM; n = 298 neurons), MGB = 67.06 +/- 3.83% (mean +/- SEM; n = 597 neurons); *Figure 1D*; *Figure 1—source data 1*). Despite the comparable thalamic effect, the impact on target cortical areas was dramatically distinct, which we assessed separately for putative excitatory neurons (regular spiking, RS) and putative inhibitory ones (fast spiking, FS; see Materials and methods for unit label identification and *Figure 1—figure supplement 1* for clustering and waveform examples). Specifically, in the PL, MD activation suppressed the spike rate of RS neurons and predominantly enhanced FS neural spiking (n = 441 RS, 265 FS neurons; group averages: *Figure 1E–H*, *Figure 1—source data 1*; individual animals: *Figure 1—figure supplement 2*, *Figure 1—figure supplement 2—source data 1*). In contrast, MGB activation robustly drove A1 RS neurons with variable effects on FS neurons (i.e. both excitatory and inhibitory effects were observed with similar proportions, n = 724 RS, 191 FS neurons; *Figure 1E–H*, *Figure 1—source data 1*; *Figure 1—figure supplement 2*, *Figure 1—figure supplement 2—source data 1*). These findings suggested that, at least with this type of broad and sustained manipulation, the MD impacts its PL targets in a manner distinct from those observed across sensory thalamocortical loops, consistent with previous findings (*Schmitt et al., 2017*).

### MD$^{\rightarrow PL}$ projections utilize smaller synaptic terminals than MGB$^{\rightarrow A1}$ projections

To begin identifying whether the functional divergence observed across the MD$^{\rightarrow PL}$ and MGB$^{\rightarrow A1}$ pathways may be partly explained by gross structural features (*Casas-Torremocha et al., 2019*; *Rodriguez-Moreno et al., 2020*), we anterogradely labeled small populations of axons originating from the lateral MD or the ventral division of the MGB. To this end, we iontophoretically delivered biotinylated dextran amine (BDA) into these areas (*Figure 2A,D*; *Figure 2—figure supplement 1*; see Materials and methods for details).

BDA-labeled axonal varicosities (putative synaptic boutons) were Golgi-like labeled in PL (*Figure 2B,C*) or A1 (*Figure 2E,F*). To estimate the size of axonal varicosities (putative synaptic sites), maximal projection areas were measured from live images and a subset of these values were randomly picked from the two cortical areas across all layers (*Casas-Torremocha et al., 2019*). This procedure revealed a significant difference in thalamocortical bouton sizes across the MD$^{\rightarrow PL}$ and MGB$^{\rightarrow A1}$ pathways, with the latter showing a significantly higher proportion of larger boutons in

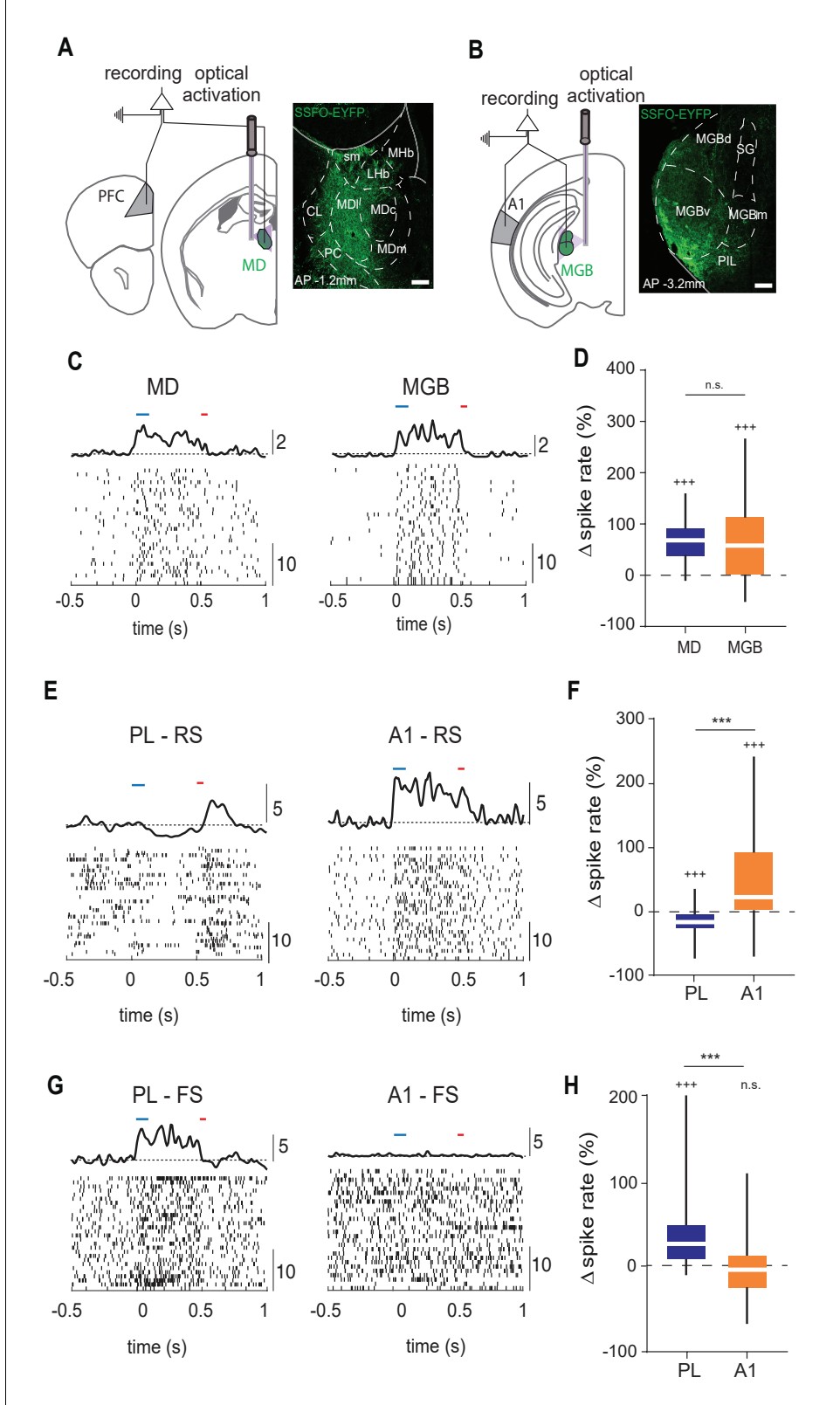

**Figure 1.** Structural differences between MD$^{\rightarrow PL}$ and MGB$^{\rightarrow A1}$ microcircuits are reflected in functional divergence. (**A**) Left: Experimental schematic of combined MD and PL recordings with optical activation of MD using Stabilized Step Function Opsins (SSFO). Right: Anatomical expression of AAV-SSFO-GFP in MD. Scale bar = 200 μm (**B**) Same as in (**A**) except for MGB$^{\rightarrow A1}$ loop. (**C**) Example raster and PSTH of single units recorded from MD (left) and

*Figure 1 continued on next page*

*Figure 1 continued*

MGB (right) during SSFO-induced activation of respective regions SSFO was activated by 100 ms pulse of blue laser (473 nm, blue bar) and terminated with a 50 ms pulse of red laser (633 nm, red bar). (**D**) MD (blue) and MGB (orange) neurons demonstrate comparably strong increases in spike rates with SSFO stimulation (n = 298 neurons, N = 4 mice for MD, n = 597 neurons, N = 5 mice for MGB, $^{+++}$p<0.001, compared to baseline rate, Wilcoxon signed rank test; n.s., compared to each other, Mann-Whitney U test). (**E**) Example raster and PSTH of RS neurons in the PL during SSFO-induced activation of MD (left, blue), and in A1 during SSFO-induced activation of MGB (right, orange). (**F**) SSFO-induced activation of the MD results in a decrease in spike rate in PFC-RS neurons, while that of the MGB leads to an increase in spike rate in A1-RS neurons (n = 441 PFC-RS neurons, $^{+++}$p<0.001 compared to baseline, Wilcoxon signed rank test; n = 724 A1-RS neurons, $^{+++}$p<0.001 compared to baseline, Wilcoxon signed rank test; $^{***}$p < 0.001 compared to each other, Mann-Whitney U test; N = 5 mice for each condition). (**G**) Example raster and PSTH of FS neurons in the PL during SSFO induced activation of the MD (left, blue), and in A1 during SSFO induced activation of the MGB (right, orange). (**H**) SSFO induced activation of the MD results in an increase in spike rate in PFC-FS neurons, while that of the MGB does not significantly alter the spike rate in A1-FS neurons (n = 265 PFC-FS neurons; $^{+++}$p<0.001 compared to baseline, Wilcoxon signed rank test, N = 5 mice; n = 194 A1-FS neurons; p=0.6531 compared to baseline, Wilcoxon signed rank test, N = 4 mice, $^{***}$p < 0.001 compared to each other, Mann-Whitney U test).

The online version of this article includes the following source data, source code and figure supplement(s) for figure 1:

**Source code 1.** MD PL firing rates.
**Source code 2.** MGB A1 firing rates.
**Source code 3.** MATLAB Script to calculate spike rate changes .
**Source data 1.** Change in firing rates normalised to baseline firing.
**Figure supplement 1.** Sorting of units into regular spiking (RS) and fast spiking (FS) neurons.
**Figure supplement 2.** Changes in spike rates of RS and FS neurons recorded in the PL or A1 separated by animals.
**Figure supplement 2—source data 1.** Change in RS and FS firing rates normalized to baseline separated by individual animals.

---

layer 4 (mean bouton projection areas: A1 (L4) = 1.28 ± 0.54 µm$^2$ vs. PL (L3) = 0.77 ± 0.42 µm$^2$; p < 0.001, Kruksal Wallis test with Dunn's post hoc test, *Figure 2—source data 2*; p < 0.001, Kolmogorov-Smirnov test *Figure 2G,H*; *Figure 2—source data 2*). We however noted that MD boutons in all layers of PL and in MGB boutons in layers 1, 3, and 5 of A1 are significantly smaller, than MGB boutons in layer 4 of A1. This substantial difference in bouton size is consistent with the notion that the MGB$^{→A1}$ is a driver pathway (*Casas-Torremocha et al., 2017*; *Viaene et al., 2011*), whereas the MD$^{→PL}$ pathway may be better described as a modulatory one. The MGB findings are also consistent with previous studies using a comparable approach (*Viaene et al., 2011*).

## MD$^{→PL}$ projections contact a larger proportion of PV+ inhibitory neurons compared to MGB$^{→A1}$ projections

To examine additional structural sources of functional divergence across the MD$^{→PL}$ and MGB$^{→A1}$ pathways, we took advantage of the recently characterized trans-synaptic anterograde adeno-associated virus (AAV) as a labeling tool (*Zingg et al., 2017*). Specifically, we used this tool to label, respectively postsynaptic targets of MD and MGB (thalamic injection sites, *Figure 3—figure supplement 1*) within the PL and A1 with cre-recombinase, followed by the injection of a second AAV harboring a pan-neuronal promoter driving cre-dependent expression of a fluorescent protein, mCherry (*Figure 3A*). In that manner, all neural postsynaptic targets were labeled across those two cortical areas of interest (*Figure 3B*, cortical injection site of AAV-FLEX-mCherry).

We performed this experiment across four mice for each system (MD$^{→PL}$ and MGB$^{→A1}$). Although both experiments (MD$^{→PL}$ and MGB$^{→A1}$) labeled comparable putative postsynaptic neurons across animals (*Figure 3—figure supplement 2*, *Figure 3—figure supplement 2—source data 1*), there were important differences in the spatial localization of these neurons. More specifically, while MGB targeted A1 neurons were most densely localized to L4, MD targeted PL neurons were more spatially diffuse (*Figure 3C–E*; n = 2 to 3 sections, N = 4 mice per condition, *Figure 3—source data 1*). This is consistent with previous literature showing broad differences of axonal termination across these two systems (*Kuramoto et al., 2017*; *Llano and Sherman, 2008*).

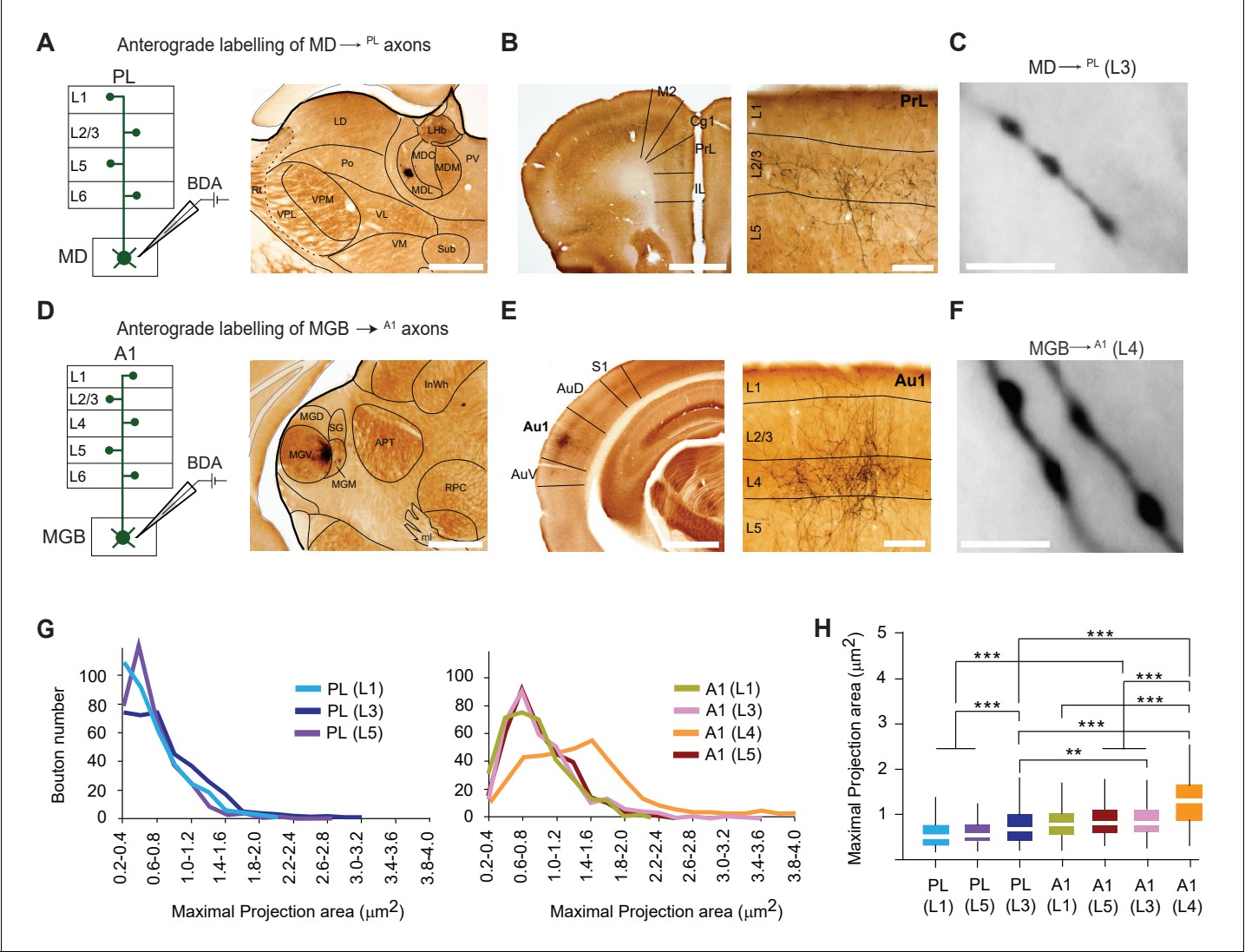

**Figure 2.** MD→PL axons establish smaller synaptic terminals than MGB→A1 projections. (**A**) Schematic diagram of the iontophoretic injection protocol (left) and images of the center of a representative BDA deposit in the lateral MD (right) Scale bar = 500 µm. (**B**) Low magnification coronal section of the PL showing areal (left) and laminar (right) distribution of BDA-labeled MD→PL axons. (**C**) High-magnification images show smaller, more frequent MD→PL boutons in layer 3 of PL. Scale bar = 5 µm. (**D**) Schematics of iontophoretic injection protocol (left) and representative image of BDA deposit in the ventral MGB (right) Scale bar = 500 µm. (**E**) Low magnification coronal section of A1 showing areal (left) and laminar (right) distribution of BDA-labeled MGB→A1 axons. (**F**) High-magnification images show larger, less frequent MGB→A1 boutons in layer 4 of A1. Scale bar = 5 µm. (**G**) Distribution of thalamocortical axonal bouton sizes (maximal projection area, in µm$^2$) in layers 1, 3 and 5 of PL (left chart) can be compared to that of bouton sizes in layers 1, 3, 4 and 5 of A1 (right chart). (**H**) Comparison of mean maximal projection areas (µm$^2$) among MD→PL boutons (layers 1, 3 and 5) and MGB→A1 boutons (layers 1, 3, 4 and 5). (***$p < 0.001$; Kruskal Wallis test with Dunn's corrected multiple comparison; See *Figure 2—source data 2*). Data are plotted from BDA labeling experiments in N = 3 mice for each nucleus where 350 varicosities were measured per cortical layer (2450 varicosities in total).

The online version of this article includes the following source data and figure supplement(s) for figure 2:

**Source data 1.** Maximal projection areas of BDA labeled boutons.
**Source data 2.** p-values table for KS test and Dunn's corrected multiple comparisons post Kruskal Wallis test in Figure 2G, H.
**Figure supplement 1.** Images of BDA deposits in lateral MD and ventral MGB.

To take further advantage of this experiment, we asked whether there was a difference in cellular composition of the postsynaptic cortical targets across these two systems. We reasoned that this could provide a link to the functional experiment described earlier (*Figure 1*; *Figure 1—figure supplement 2*). Because FS neurons are predominantly parvalbumin (PV) positive, and because primary

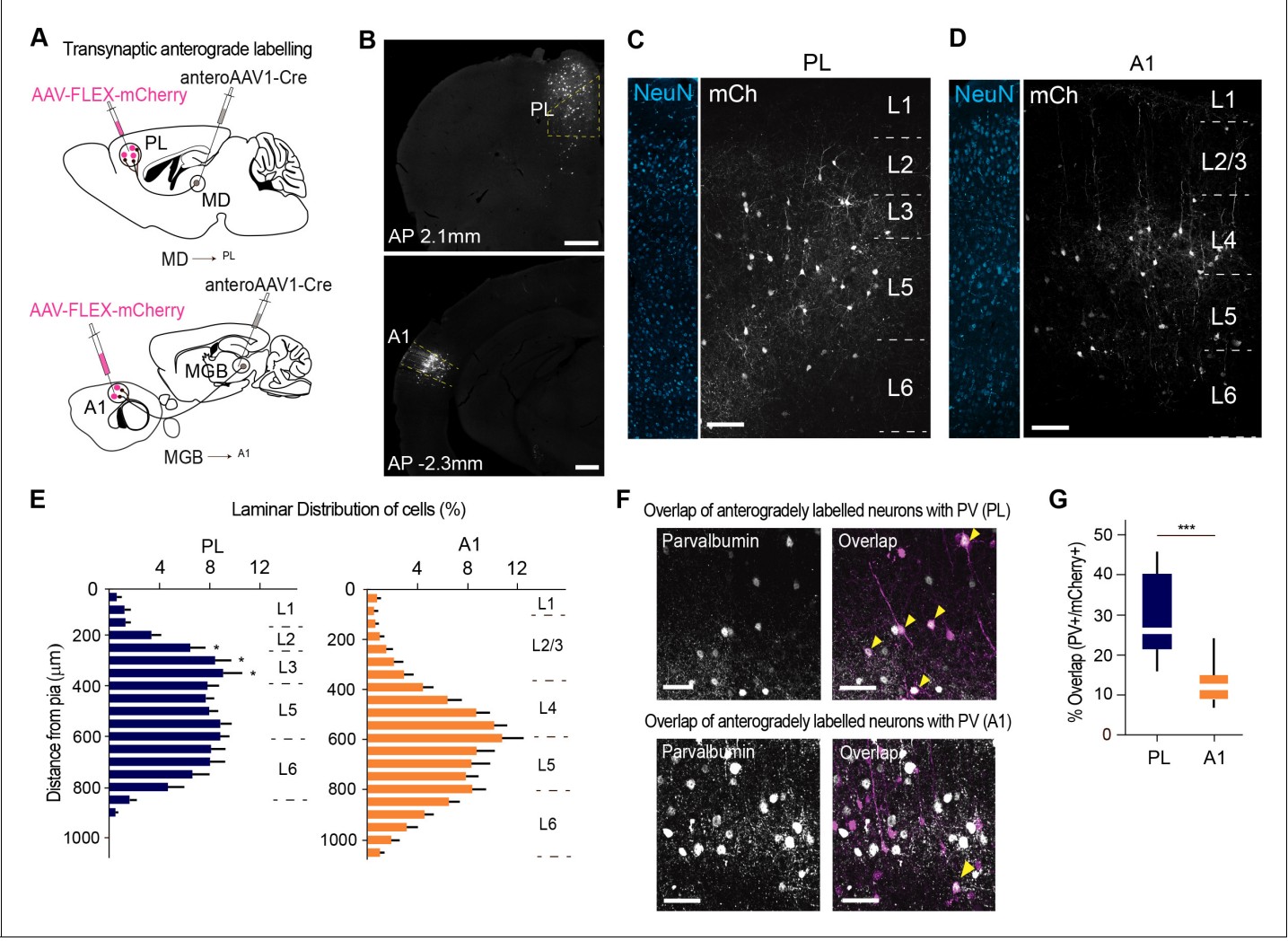

**Figure 3.** MD→PL projections contact a larger proportion of PV+ inhibitory neurons compared to MGB→A1 projections. (**A**) Schematics of trans-synaptic anterograde labeling strategies for output targets of mediodorsal (top) and geniculate (bottom) thalamocortical cells. (**B**) Representative image of PL (top) and A1 (bottom) showing specificity of cortical injection site and trans-synaptic labeling. Scale bars = 500 μm. (**C**) Representative confocal images of PL neurons trans-synaptically labeled by the MD show labeling of output neurons across layers 2 to 5. Scale bar = 100 μm. (**D**) A1 neurons trans-synaptically labeled by the MGB primarily reside in layer 4. Scale bar = 100 μm. (**E**) Laminar distribution of trans-synaptically labeled cells in PL (left, blue) versus A1 (right, orange) quantitatively confirm the MD's projections to PL are uniformly expressed across layers 2, 3, and 5, while the MGB's projections to A1 mainly target layer 4. (***p<0.0001, Kruskal-Wallis test with Dunn's corrected multiple comparison; p values for the significant differences are (from top to bottom) *p=0.0129, *p=0.0159, *p=0.0206) (**F**) Representative high-magnification images of immunohistochemically labeled parvalbumin neurons in PL (top panel, left), and their overlap with mCherry+ MD output neurons (top panel, right). Representative high magnification images of immunohistochemically labeled parvalbumin neurons in A1 (left) and their overlap with mCherry+ MGB output neurons (right) are shown in the bottom panel. Yellow arrowheads indicate co-labeled neurons. Scale bars = 50 μm. (**G**) Quantification of percentage of anterogradely labeled neurons which co express PV reveal significantly higher MD innervation of PV interneurons in the PL (blue) than MGB innervation to PV interneurons in A1 (orange) (***p=0.0004, Mann Whitney U test). Data is plotted from n = 3 sections per mouse and N = 4 mice for each condition.

The online version of this article includes the following source data and figure supplement(s) for figure 3:

**Source data 1.** Laminar distribution of transsynaptic anterograde labeling and overlap with cortical PV+ neurons .
**Figure supplement 1.** Schematics and injection sites of AAV1 mediated anterograde tracing.
**Figure supplement 2.** Anterograde transsynaptic labeling distributions in PL and A1.
**Figure supplement 2—source data 1.** Quantification of number of transynaptic anterograde neurons in PL vs A1.
**Figure supplement 3.** Immunolabeled PV positive neuron distribution in PL and A1.

sensory thalamic circuits are known to predominantly target this type of inhibitory neurons (*de la Rocha et al., 2008*), we asked whether there were quantitative differences in the number of cortical PV interneurons targeted by MD$^{\rightarrow PL}$ vs. MGB$^{\rightarrow A1}$ neurons.

Staining for PV followed by post-hoc analysis of the fraction of anterogradely labeled trans-synaptic neurons that co express PV (co-labeled/mCherry+ neurons), showed that the MD$^{\rightarrow PL}$ targets approximately twice as many PV+ interneurons as the MGB$^{\rightarrow A1}$ pathway does (*Figure 3F,G*; n = 2 to 3 sections per animal; N = 4 mice per condition) while the density of PV+ neurons in the PL and A1 remain comparable (*Figure 3—figure supplement 3*, *Figure 3—figure supplement 2—source data 1*). Our results are in line with previous results indicating substantial inhibitory innervation of A1 PV neurons by MGB afferents, (*de la Rocha et al., 2008*), but additionally indicate that the proportional innervation of these inhibitory neurons in the PL by thalamic afferents is even larger.

## MGB$^{\rightarrow A1}$ neurons innervate excitatory cortical neurons with larger synaptic terminals compared to MD$^{\rightarrow PL}$ neurons

At this point, our anterograde experiments pointed to two major differences in cortical innervation across the MD$^{\rightarrow PL}$ and MGB$^{\rightarrow A1}$ pathways: terminal sizes (*Figure 2*) and E/I innervation (*Figure 3*). Are these structural variations related to one another in some manner? Meaning, are there differences in terminal sizes that also relate to the identity of the cortical postsynaptic target? This question is of particular relevance given that previous literature suggested the existence of high efficacy thalamocortical transmission across sensory pathways in the form of significant short-latency cross-correlations across functionally aligned LGN V1 neural pairs (*Alonso et al., 1996*; *Alonso et al., 2001*).

To address this question, we implemented the mammalian GFP reconstitution across synaptic partners (mGRASP) technique. This approach is based on functional complementation between two nonfluorescent GFP fragments that are localized to a pre- and postsynaptic pair, thereby specifically labeling the synapses between two targets for more accurate quantification (*Feng et al., 2014*). In fact, recent studies have shown excellent correlation between the size of synapses detected using mGRASP and synaptic strength measured electrophysiologically (*Song et al., 2018*). Because this method primarily revealed GFP+ structures at the somas of postsynaptic neurons as had been shown in previous studies (*Song et al., 2018*), we quantified these structures as a correlate for endogenous synaptic innervation (see Discussion).

To label MD$^{\rightarrow PL}$ and MGB$^{\rightarrow A1}$ axosomatic synapses with mGRASP, we transduced MD or MGB with an AAV harboring pre mGRASP:mCerulean and their postsynaptic targets in the PL and A1 (*Figure 4A*), respectively, with an AAV harboring a cre-dependent post-mGRASP:TdTomato construct. These experiments were performed across both excitatory (CamKII+, n = 40 individual neurons, N = 4 mice for each condition) and inhibitory (PV+, n = 40 individual neurons, N = 4 mice for each condition) neurons to separately evaluate synaptic terminals onto the two components of E/I balance (*Figure 4B,C*). Consistent with previous results in rodent cortex (*Smith and Populin, 2001*), the predominant layer four thalamorecepient excitatory cell type is pyramidal (*Barbour and Callaway, 2008*; *Sakata and Harris, 2009*). PV+ postsynaptic neurons were targeted using a transgenic mouse line expressing cre recombinase under the control of the PV promoter. Postsynaptic CamKII+ neurons, on the other hand, were targeted using an AAV expressing cre recombinase from a CamKII promoter co-injected along with the cre-dependent AAV harboring post-mGRASP:TdTomato (*Figure 4B,C*).

We found that although the MD$^{\rightarrow PL}$ made more mGRASP labeled synapses across the two neuronal types (Excitatory and Inhibitory; *Figure 4D*, *Figure 4—source data 1*), the terminal sizes were strikingly different for those impinging on cortical excitatory (CamKII+) neurons, compared to the MGB$^{\rightarrow A1}$ pathway. More specifically, MGB neurons innervated excitatory A1 neurons with significantly larger synaptic terminals, an effect that was specific to this neural population because we did not detect any differences in the sizes of thalamocortical terminals onto PV+ inhibitory neurons across the two systems (*Figure 4E*, *Figure 4—source data 1*). As a further validation, we used immunohistochemistry to verify the thalamocortical origins of the synapses labeled by mGRASP. Specifically, we verified the preferential colocalization of the mGRASP signal with a thalamocortical synaptic marker, VGluT2 over VGluT1 (*Nahmani and Erisir, 2005*), which labels synapses of non-thalamic origin (*Figure 4—figure supplement 1*, *Figure 4—figure supplement 1—source data 1*). Overall, these results provide a structural basis for the notion that the MGB$^{\rightarrow A1}$ is a driver thalamocortical pathway at a single-cell resolution.

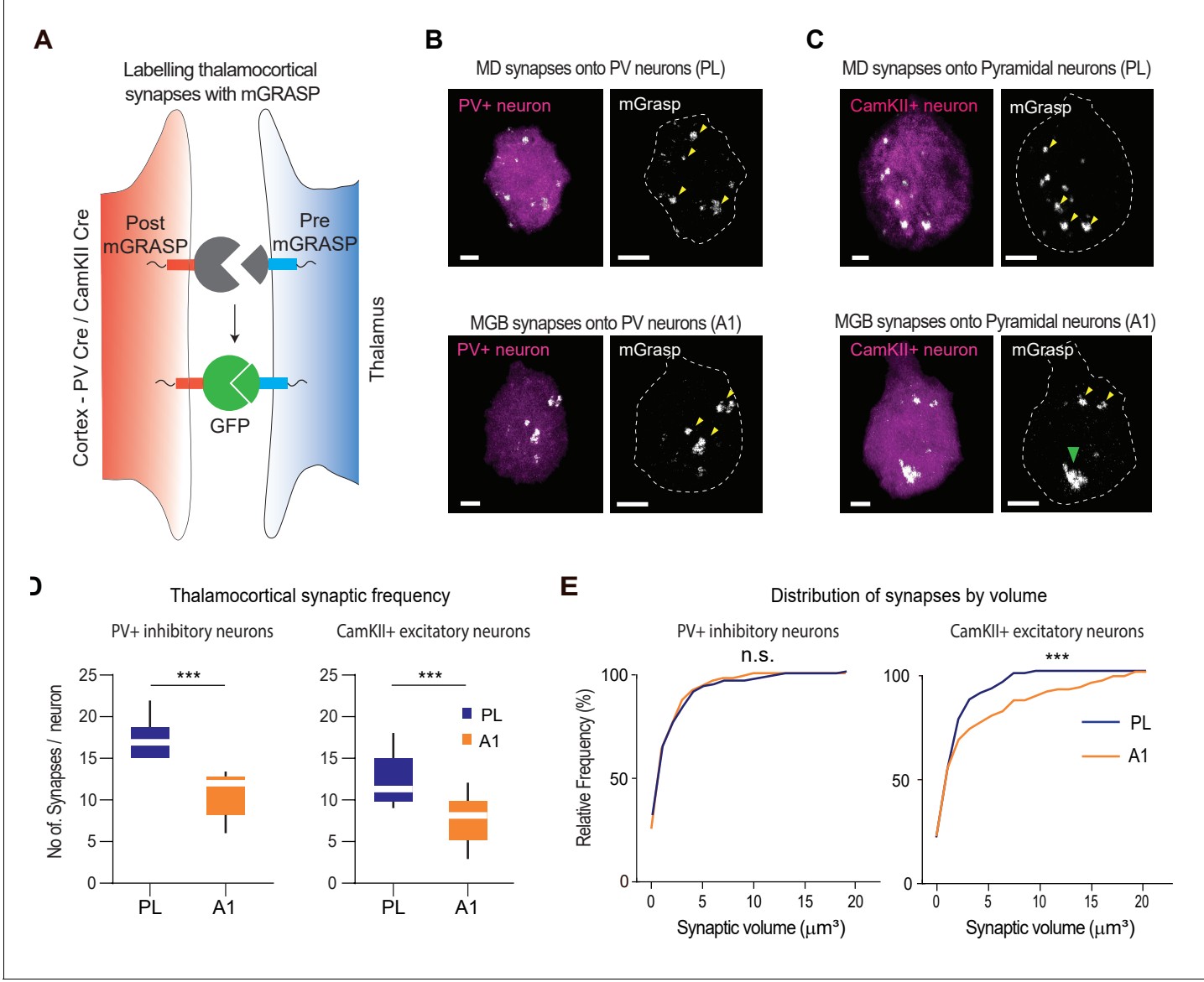

**Figure 4.** MGB$^{\rightarrow A1}$ neurons innervate excitatory cortical neurons with larger synaptic terminals compared to MD$^{\rightarrow PL}$ neurons. (**A**) Schematic of synaptic labeling mechanism using PV-Cre driver lines or CamKII-Cre virus to label inhibitory or excitatory synapses, respectively. (**B**) High magnification confocal image of a post-mGRASP labeled PV+ neuron (magenta) in the PL (top) and A1 (bottom) merged with synaptic labeling by mGRASP (white). Arrowheads indicate putative synapses. Scale bars = 3 μm. (**C**) High magnification confocal images of a post-mGRASP labeled CamKII+ neuron (magenta) in the PL (top) and A1 (bottom) merged with synaptic expression from mGRASP (white). Scale bars = 3 μm. (**D**) Quantification of thalamocortical synaptic frequency in PV+ inhibitory neurons (left), and CamKII+ excitatory neurons (right), shows the PL (blue) receives significantly higher frequency of thalamocortical synapses than A1 (orange). (***p<0.0001 for PV neurons, ***p<0.0001 for Pyramidal neurons, Mann Whitney U test). (**E**) Left: Cumulative frequency distribution of synapses by volume in PV+ inhibitory neurons show no significant difference of volume between the thalamocortical synapses in the PL (blue) and A1 (orange) (n.s., p=0.1703, Kolmogorov-Smirnov test). Right: Cumulative frequency distribution of synapses by volume in CamKII+ excitatory neurons show a significant difference in synaptic volume between thalamocortical synapses in PL (blue), and A1 (orange), with larger synapses on A1 excitatory neurons from MGB. (***p=0.0007, Kolmogorov-Smirnov test). Data is plotted from n = 40 individual PV+ and CamKII+ neurons each from N = 4 mice for each condition.

The online version of this article includes the following source data and figure supplement(s) for figure 4:

**Source data 1.** Synaptic counts and volumes of mGRASP labeled synapses.

**Figure supplement 1.** Overlap of mGRASP labeled synapses with vGlut1 and vGlut2.

**Figure supplement 1—source data 1.** Quantification of overlap of mGRASP labeled synapses with vGluT1 and vGluT2.

## MD$^{\rightarrow PL}$ neurons receive a diverse set of monosynaptic cortical inputs

To further characterize the structural differences across the two thalamocortical loops, we sought to map the inputs of the MD$^{\rightarrow PL}$ and MGB$^{\rightarrow A1}$ microcircuits with higher specificity. As such, and to precisely identify the set of monosynaptic inputs innervating MD$^{\rightarrow PL}$ neurons, we used a two-step labeling approach. First, we rendered MD$^{\rightarrow PL}$ neurons susceptible to monosynaptic rabies tracing by injecting a retrograde adeno-associated virus (AAVrg) harboring cre-recombinase (*Tervo et al., 2016*) into the PL and two helper AAV viruses, one of which is cre-dependent to achieve pathway specificity, directly into the MD. Second, and following 2 weeks of incubation, we injected G-deleted pseudorabies into the MD (*Figure 5—figure supplement 1*; see Materials and methods) (*Chatterjee et al., 2018*). Consistent with known connectivity patterns between the MD and PL, we found that the majority of cre expressing MD$^{\rightarrow PL}$ neurons were in the lateral MD subdivision. This also ensured that starter neurons within this subpopulation were predominantly localized to that subdivision (*Figure 5—figure supplement 2*, *Figure 5—figure supplement 2—source data 1*). We compared the results from this experiment with those obtained from a similar approach to the MGB$^{\rightarrow A1}$ pathway (*Figure 5—figure supplements 1* and *2*).

*Figure 5* (AP 2.60 to −0.80 mm from Bregma) and 6 (AP −2.00 to − 5.00 mm from Bregma) show coronal sections throughout the rostro-caudal axis of the brain, spanning both the forebrain and midbrain. Visual inspection of these representative sections showed that the MD$^{\rightarrow PL}$ population receives substantial projections from frontal cortical areas including the prelimbic, cingulate and secondary motor cortices (*Figure 5A–C*). In contrast, MGB$^{\rightarrow A1}$ neurons receive comparatively little cortical input, compared to their known driving inputs from inferior colliculus (*Figure 6E*), which we also find through this method of monosynaptic rabies tracing. Beyond these major differences, the MD$^{\rightarrow PL}$ neurons, compared to MGB$^{\rightarrow A1}$ neurons, receive stronger inputs from motor related subcortical structures such as the deep layers of the superior colliculus, substantia nigra pars reticulata, lateral hypothalamus, zona incerta, ventral pallidum, lateral preoptic area, and anterior olfactory area (*Figure 6*).

Quantitative analysis across several mice (N = 5 mice per condition) confirmed these representative data (*Figure 5—figure supplement 3*, *Figure 5—figure supplement 3—source data 1*). That is, the largest source of extrathalamic inputs to MD$^{\rightarrow PL}$ was cortical (*Figure 5—figure supplement 3A*), while that to the MGB$^{\rightarrow A1}$ was from the midbrain (*Figure 5—figure supplement 3B*). Importantly, we were able to make quantitative comparisons between these circuits on a single-cell basis, by normalizing these inputs to the number of starter cells seen in the MD or MGB of individual animals. This analysis suggests that when compared on a single neuron basis MD$^{\rightarrow PL}$ neurons receive five times as many cortical inputs relative to individual MGB$^{\rightarrow A1}$ neurons (*Figure 7A,B*, *Figure 7—source data 1*), while the opposite pattern holds for midbrain inputs (*Figure 7—figure supplement 1*). Furthermore, we also found that the cortical inputs to MD$^{\rightarrow PL}$ showed unique layer-wise distributions where inputs from orbitofrontal cortex (OC) and PL were predominantly from layer 5 (*Figure 7C*) while inputs from secondary motor cortex (M2) and cingulate cortex (Cg) were from layer 6 (*Figure 7C*). In contrast cortical inputs to MGB$^{\rightarrow A1}$ were by and large restricted to layer 6 (*Figure 7D*) with sparse input neurons present in layer 5 (*Figure 7C*). We also found that projections within the thalamus to these two thalamic populations were limited to the thalamic reticular nucleus (TRN) as is known from previous studies (*Pinault, 2004*). Importantly, these projections were from distinct TRN subnetworks as would be predicted from more recent anatomical and functional studies (*Figure 7—figure supplement 1*; *Clemente-Perez et al., 2017*; *Crabtree, 2018*; *Halassa et al., 2014*; *Krol et al., 2018*). Observation of these distinct TRN inputs also served to confirm the specificity of monosynaptic tracing strategy to map brain wide inputs to MD$^{\rightarrow PL}$ and MGB$^{\rightarrow A1}$ thalamocortical neurons.

## Discussion

The incomplete information available on input/output connectivity motifs across thalamocortical loops is a major obstacle to our understanding of the thalamus and its role in cortical function. Here, we compared the input/output (I/O) connectivity patterns of an associative thalamocortical loop with that of a sensory thalamocortical loop. Specifically, for an exemplar cognitive corticothalamic loop we explored the I/O connectivity of mediodorsal thalamic neurons which innervate the prelimbic area of the prefrontal cortex (MD$^{\rightarrow PL}$) and are predominantly in the lateral part of the MD. On the

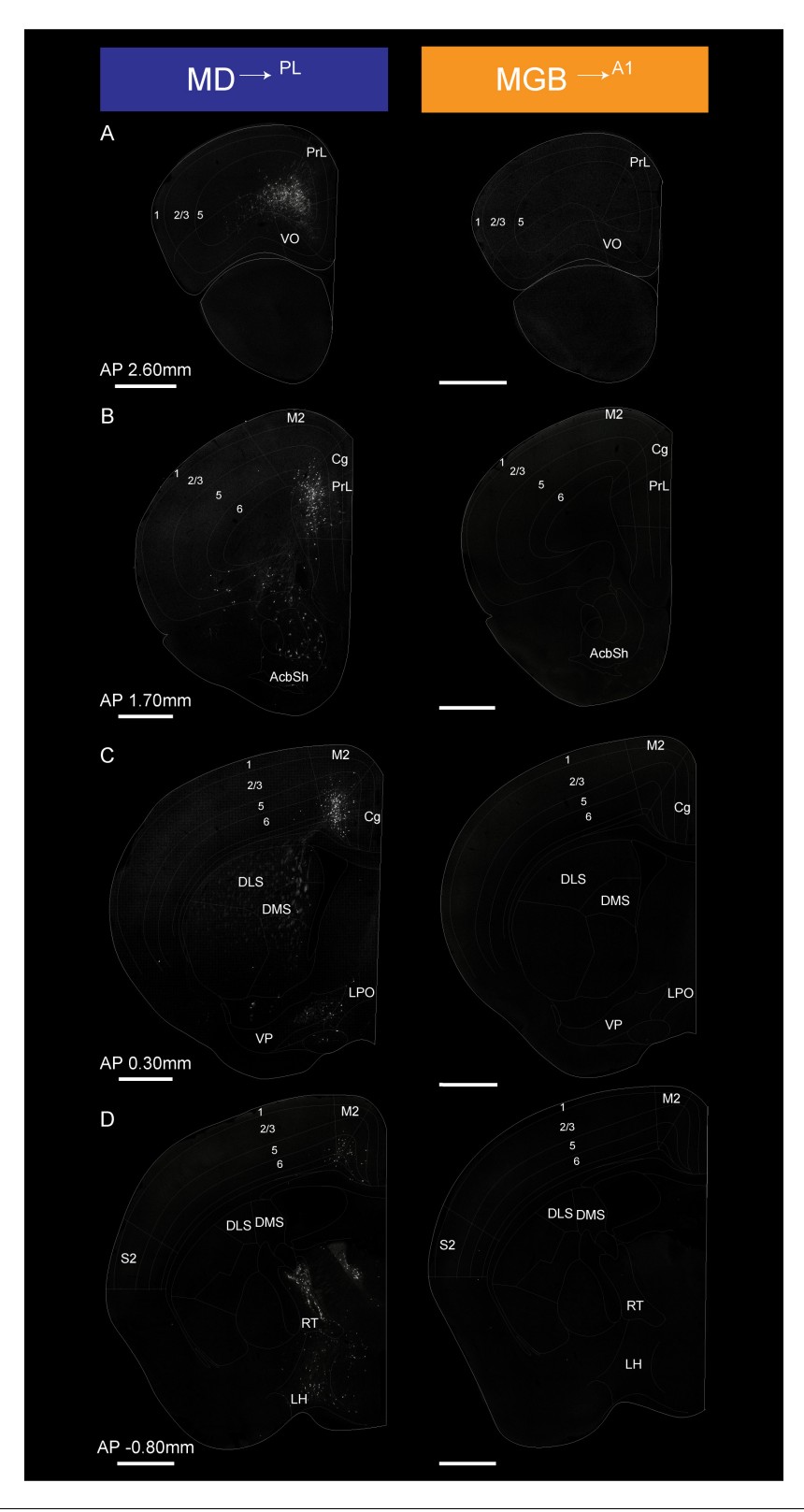

**Figure 5.** Detailed map of retrograde inputs to geniculate microcircuit compared to mediodorsal microcircuit (Anterior half, AP 2.60 mm to −0.80 mm). (A–D) Representative confocal images of monosynaptic inputs MD$^{\rightarrow PL}$ neurons (left) and MGB$^{\rightarrow A1}$ neurons (right) across the rostro-caudal axis illustrate distinct input patterns in each microcircuit. Distance from bregma appears between each pair of images. N = 5 mice for each condition. Scale bars = 1 mm.

*Figure 5 continued on next page*

*Figure 5 continued*

The online version of this article includes the following source data and figure supplement(s) for figure 5:

**Figure supplement 1.** Monosynaptic retrograde tracing from MD$^{\rightarrow PL}$ and MGB$^{\rightarrow A1}$ neurons.
**Figure supplement 2.** Distribution of starters for monosynaptic retrograde tracing from MD$^{\rightarrow PL}$ and MGB$^{\rightarrow A1}$ neurons.
**Figure supplement 2—source data 1.** Starter counts for monosynaptic retrograde tracing with rabies viruses.
**Figure supplement 3.** Quantitative summary of retrograde monosynaptic input tracing from MD$^{\rightarrow PL}$ and MGB$^{\rightarrow A1}$ neurons.
**Figure supplement 3—source data 1.** Retrograde inputs to MD vs MGB expressed as a percentage of total inputs.

other hand, for an exemplar primary sensory thalamocortical loop we investigated the I/O connectivity of the medial geniculate nucleus neurons that project to the primary auditory cortex (MGB$^{\rightarrow A1}$) and are located in the ventral division of MGB. The analysis reveals several interesting differences between the fine circuit organization of these two thalamocortical pathways.

First, there are major differences in the effect of MGB$^{\rightarrow A1}$ and MD$^{\rightarrow PL}$ outputs upon their respective cortical targets. Compared to the MGB which robustly drives excitation of A1, equivalent MD activation results in a mild suppression of PL excitatory neurons. This is contrasted by the strong enhancement of fast spiking interneuron activity within this associative cortical area. Recent evidence supports a central role of these fast-spiking and presumably PV+ interneurons in regulating prefrontal E/I balance and consequent behavioral flexibility (*Cho et al., 2015*; *Kim et al., 2016*; *Kvitsiani et al., 2013*). Our anterograde tracing data show that the MGB$^{\rightarrow A1}$ neurons preferentially target neurons in layer 4 of the auditory cortex through large and less frequent driver type synapses (*Llano and Sherman, 2008*; *Nahmani and Erisir, 2005*; *Raczkowski and Fitzpatrick, 1990*; *Smith et al., 2012*), providing a structural basis for the aforementioned functional effects. In contrast, MD$^{\rightarrow PL}$ neurons targets neurons across multiple layers of the PL via frequent and smaller synaptic boutons suggesting a modulatory effect of MD neurons onto their cortical targets (*Mease et al., 2017*; *Rikhye et al., 2018b*). Notably and in line with a previous study (*Delevich et al., 2015*), the MD$^{\rightarrow PL}$ neurons also make more frequent contacts with local inhibitory parvalbumin+ (PV+) neurons in the prefrontal cortex compared to MGB$^{\rightarrow A1}$ neurons in the A1. However, we should also emphasize that this is not the only functional effect observed. Previous studies, including our own, indicate that associative thalamic drive enhances lateral effective connectivity in cortex required for maintaining working memory and attentional control (*Schmitt et al., 2017*). In fact, the main difference observed across these two pathways using BDA labeling (*Figure 2*) appears to be accounted for by excitatory, not inhibitory innervation, based on the mGRASP experiment (*Figure 4*). That, coupled with the finding from trans-synaptic anterograde labeling, that the MD$^{\rightarrow PL}$ output targets a larger fraction of inhibitory neurons than the MGB$^{\rightarrow A1}$ does, strongly suggests that the functional impact of activating these circuits on their cortical targets may be distinct, perhaps with differences in the resulting E/I balance (*Yizhar et al., 2011*; *Delevich et al., 2020*; *Fan and Hu, 2018*; *Ferguson and Gao, 2018*).

Second, our monosynaptic retrograde tracing study showed that the cortical inputs to the MD$^{\rightarrow PL}$ neurons are the largest source of monosynaptic extrathalamic inputs and originate from both, layers 5 and 6 of the frontal cortex. These observations are in stark contrast to the origin of monosynaptic inputs to the MGB$^{\rightarrow A1}$. Here, the cortical inputs constitute relatively, a smaller component of the total inputs and are restricted to layer 6 of the auditory cortex. Thus, the MGB$^{\rightarrow A1}$ neurons are reflective of a classical thalamic relay (*Sherman and Guillery, 1998*; *Briggs and Usrey, 2011*). Their main inputs originate from the main midbrain relay nucleus of the auditory pathway, the inferior colliculus, while the cortex contributes a smaller input which originates from layer 6. Notably, while the primary inputs to the MD are cortical in origin, in line with it being an associative thalamic structure, they are not exclusive to layer 5. In fact, the largest cortical input to MD$^{\rightarrow PL}$ neurons originate primarily from layer 6 of secondary motor cortex (M2) and the cingulate cortex (Cg).

Third, we show that the cortical inputs to individual MD$^{\rightarrow PL}$ neurons, as measured by input to starter ratios, are three to five times more numerous than those to MGB$^{\rightarrow A1}$ neurons. Moreover, the cortical inputs to MD$^{\rightarrow PL}$ neurons predominantly originate from either layer 5 or layer 6 of a variety of frontal cortical areas beyond the prelimbic cortex. These areas have been shown to have diverse functions like motor planning (area M2) (*Chen et al., 2017*; *Guo et al., 2017*), hierarchical decision making (area Cg) (*Sarafyazd and Jazayeri, 2019*) and value-based decision making (area OC)

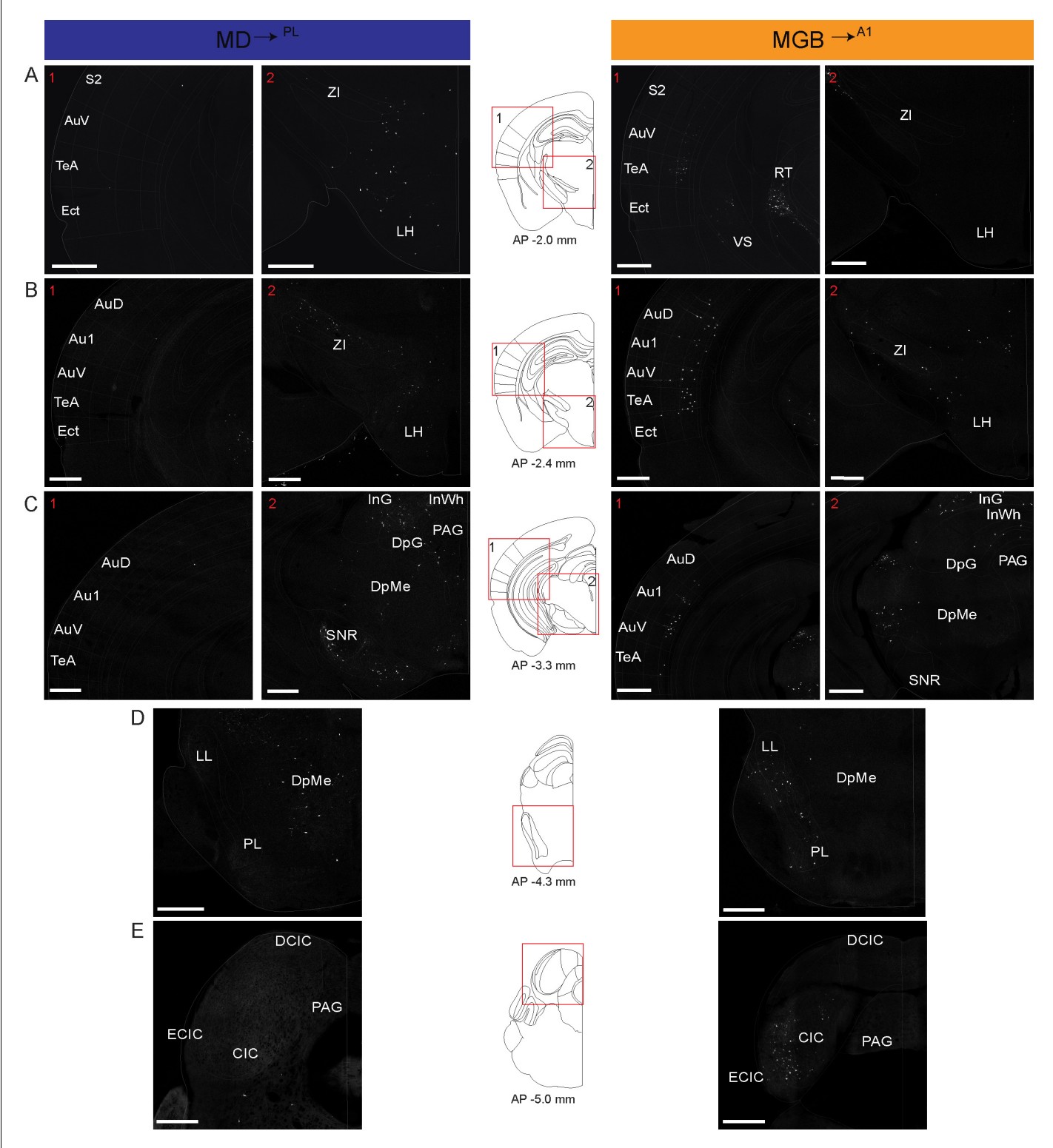

**Figure 6.** Detailed map of retrograde inputs to geniculate microcircuit compared to mediodorsal microcircuit (Rostral half, AP 2.60 mm to −0.80 mm). (A–E) Representative confocal images of monosynaptic inputs to inputs MD$^{→PL}$ neurons (left) and MGB$^{→A1}$ neurons (right) across the rostro-caudal axis illustrate distinct input patterns in each microcircuit. Distance from bregma appears beneath the corresponding unified anatomical atlas (**Chon et al., 2019**) section shown (middle). Regions showed are depicted as numbered (**A–C**) red boxes on each corresponding atlas section for reference. N = 5 mice for each condition. Scale bars = 500 μm.

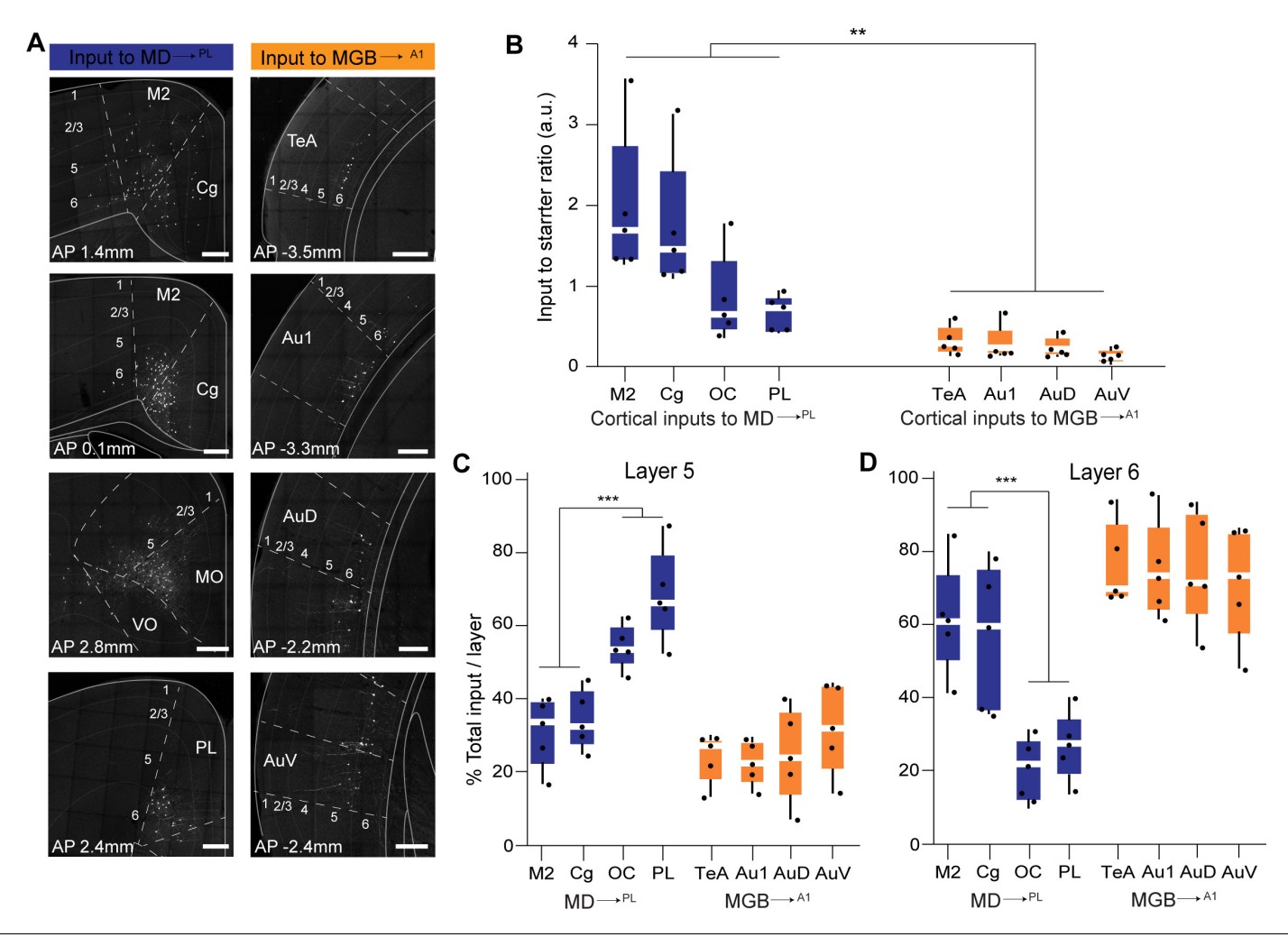

**Figure 7.** Representation and quantification of inputs to the mediodorsal microcircuit compared to the geniculate microcircuit. (**A**) Representative images of major cortical inputs to MD$^{\rightarrow PL}$ loop (left) versus MGB$^{\rightarrow A1}$ (right). Scale bars = 200 µm. (**B**) Quantification of input to starter ratios for each major cortical input region to MD$^{\rightarrow PL}$ neurons (left, blue) versus MGB$^{\rightarrow A1}$ neurons (right, orange) reveal significantly denser cortical inputs per thalamic neuron in the MD$^{\rightarrow PL}$ loop compared to the MGB$^{\rightarrow A1}$ loop (**\*\*p=0.0079; Mann Whitney U test with all cortical inputs to MD$^{\rightarrow PL}$ neurons treated as one group and compared to all cortical inputs to MGB$^{\rightarrow A1}$ neurons). Each solid black dot corresponds to one data point that is one animal (**C**) Contribution of layer five neurons to the net cortical input from each major cortical region to MD$^{\rightarrow PL}$ neurons (left, blue) versus MGB$^{\rightarrow A1}$ neurons (right, orange) reveal significantly higher layer five inputs from the OC and PL than those from M2 and Cg to the MD$^{\rightarrow PL}$ neurons (**\*\*p<0.0001; Mann Whitney U test with all layer 5 cortical inputs to MD$^{\rightarrow PL}$ neurons arising from M2 and Cg treated as one group and compared to all cortical to MD$^{\rightarrow PL}$ neurons arising from OC and PL). Layer 5 inputs to MGB$^{\rightarrow A1}$ neurons constitute only a minor fraction of their cortical inputs. Each solid black dot corresponds to one data point i.e. one animal. (**D**) Contribution of layer six neurons to the net cortical input from each major cortical region to MD$^{\rightarrow PL}$ neurons (left, blue) versus MGB$^{\rightarrow A1}$ neurons (right, orange) reveal a predominant layer 6 input from all major cortical input regions to the MGB$^{\rightarrow A1}$ loop. Of all major cortical inputs to MD$^{\rightarrow PL}$ neurons, the M2 and Cg provide significantly higher layer six input than the OC and PL (**\*\*\*p=0.0004; Mann Whitney U test with all layer five cortical inputs to MD$^{\rightarrow PL}$ neurons arising from M2 and Cg treated as one group and compared to all cortical to MD$^{\rightarrow PL}$ neurons arising from OC and PL). Each solid black dot corresponds to one data point that is one animal. Data is plotted from N = 5 mice for each condition. The online version of this article includes the following source data and figure supplement(s) for figure 7:

**Source data 1.** Retrograde monosynaptic labeling input counts, input to starter ratios and layerwise distribution of inputs.

**Figure supplement 1.** Mid brain to starter ratios and TRN inputs to MD$^{\rightarrow PL}$ and MGB$^{\rightarrow A1}$ neurons.

(**Padoa-Schioppa and Conen, 2017**; **Rich and Wallis, 2016**). The thalamus, unlike the cortex, lacks local excitatory recurrent connections. Therefore, the computations performed by thalamic neurons depend on its long-range excitatory inputs (**Halassa and Kastner, 2017**). Considering this fact, the diversity of cortical inputs to the MD suggests that MD neurons might perform integrative

computations on cortical inputs carrying different cognitive control signals instead of relaying them directly to the cortex. This provisional conclusion is supported by a recent finding which showed that thalamic neurons in the MD represent conjunctions of cortical signals to encode contextual information (*Rikhye et al., 2018a*). Further studies of input connectivity at the level of single cells need to be performed to see if there is convergence of cortical inputs onto individual MD neurons as has been inferred for another associative thalamic nuclei, the Pulvinar, from functional studies in rodents as well as non-human primates (*Jaramillo et al., 2019*; *Komura et al., 2013*).

It is worth noting that our study is not without limitations. For example, it is well-known that most endogenous thalamocortical inputs innervate the dendritic compartments instead of the soma. Thus, we interpret the mGRASP experiments as a correlate for these endogenous inputs rather than a ground-truth reflection as would be revealed by immunohistochemistry for example. In addition, the rabies tracing results may suffer from both false positive and negative inputs, along with the known underestimation of starter population spread due to rabies-induced cell death (*Lavin et al., 2020*; *Wickersham et al., 2007*). This may be one explanation for why we find sporadic labeling of thalamic nuclei in our dataset when it is known that the thalamus lacks lateral excitatory connectivity. Lastly, although AAV1 has been shown to be predominantly harbor anterograde transsynaptic properties, potential retrograde properties may be considered a confound (*Zingg et al., 2017*). However, the robust correspondence between presynaptic thalamic boutons and postsynaptic cortical cells labeled indicates that such confound has limited impact on the interpretation of the anterograde labeling data.

The original formulation of first vs. higher order thalamic nuclei is primarily related to how neurons within a thalamic nucleus are driven (i.e. whether their inputs are from sensors or cortex; *Sherman and Guillery, 1998*). The main focus of our work here is comparing a sensory and associative thalamic structure with respect to their structural outputs, along with the functional impact this structure exerts on cortical targets. From this perspective, our findings and overall synthesis may be more in line with output-based classifications such as core vs. matrix (*Jones, 1998*; *Jones, 2001*). Nonetheless, the fact that our rabies tracing shows more robust cortical inputs to the MD than the MGB is in line with its formulation as a higher order nucleus. However, a classification scheme that incorporates both inputs and outputs of single thalamic neurons would be most helpful in constraining functional models in the future (*Halassa and Sherman, 2019*).

In summary, our study supports the notion that, unlike sensory thalamic loops like MGB$^{\rightarrow A1}$, associative thalamic loops like the MD$^{\rightarrow PL}$ are composed of different input-output microcircuits residing within the same nucleus. In the future, mapping the diversity of microcircuits within cognitive thalamic nuclei, like the MD, using techniques such as high-resolution single-cell terminal mapping (*Lichtman et al., 2008*; *Peng et al., 2020*) or monosynaptic retrograde tracings from single thalamic neurons (*Ghanem and Conzelmann, 2016*; *Schubert et al., 2019*) will be necessary. The circuit motifs so revealed will help inform and generate computational models of the role of the cognitive thalamus, beyond that of a relay, in generating behavioral flexibility. In turn these circuit motifs may provide us with specific therapeutic targets for the treatment of cognitive deficits, as seen in neuropsychiatric disorders (*Mukherjee et al., 2019*; *Nakajima et al., 2019a*; *Schmitt and Halassa, 2017*).

# Materials and methods

**Key resources table**

| Reagent type (species) or resource | Designation | Source or reference | Identifiers | Additional information |
|---|---|---|---|---|
| Antibody | Anti-GFP antibody (chicken polyclonal) | Aves Labs | GFP1010 RRID:AB_2307313 | (1:1000) |
| Antibody | Alexa Fluor 488 goat anti-chicken IgG (goat polyclonal) | Thermo Fisher Scientific | A32931 RRID:AB_2762843 | (1:500) |
| Antibody | Rabbit anti-PV (rabbit polyclonal) | Swant | PV-27 RRID:AB_2631173 | (1:1000) |
| Chemical compound, drug | BDA (10 KDa) | Thermo Fisher Scientific | D1956 | 3% solution |

*Continued on next page*

*Continued*

| Reagent type (species) or resource | Designation | Source or reference | Identifiers | Additional information |
|---|---|---|---|---|
| Chemical compound, drug | Hoechst 33342 | Thermo Fisher Scientific | H3570 | 1:1000 solution |
| Software, algorithm | ImageJ | NIH | | |
| Software, algorithm | Imaris | Oxford Instruments | | Version 9.3.2 |
| Software, algorithm | Prism 8 | Graphpad | | Version 8.0 |
| Other | EnvA-RVdG-mCherry | Gift; Dr. Ian Wickersham, MIT *Chatterjee et al., 2018* | | Rabies virus expressing mCherry |
| Other | AAV1-TREtight-B19G | Gift; Dr. Ian Wickersham, MIT *Chatterjee et al., 2018* | | AAV expressing B19 variant of G protein |
| Other | AAV1-syn-FLEX-TA-TVA-GFP | Gift; Dr. Ian Wickersham, MIT *Chatterjee et al., 2018* | | AAV expressing cre dependent TVA, tTA, and GFP |
| Other | AAV2/8-CAG-pre-mGRASP-mCerulean | Gift; Dr. Michael Baratta, Univ. of Colorado | N.A. | AAV expressing pre-mGRASP and mCerulean |
| Other | AAV2/8.CAG.Jx-rev. post-mGRASP-2A-dTomato | Gift; Dr. Michael Baratta, Univ. of Colorado | N.A. | AAV expressing cre dependent post-mGRASP and tdTomato |
| Other | AAVrg-hSyn-Cre-WPRE-hGH (retrograde) | Addgene Vector Core | Lot#: 105553-AAVrg | AAV expressing cre recombinase |
| Other | AAV1-hSyn-Cre-WPRE-hGH | Addgene Vector Core | Lot#: 105553-AAV1 | AAV expressing cre recombinase |
| Other | AAV1-pCAG-FLEX-tdTomato-WPRE | Addgene Vector Core | Lot# 51503 | AAV expressing cre dependent tdTomato |
| Other | AAV1-CamKIIa-SSFO-GFP | UNC vector core | N.A. | AAV expressing SSFO and GFP under CamKII promoter |
| Other | pENN-AAV-CamKII-0.4.-Cre-SV40 | Addgene Vector Core | Lot#: 105540 | AAV expressing cre under CamKII promoter |

## Animals

A total of 43 mice were used in this study. Adult C57Bl/6 (WT) mice aged 8–12 weeks old were purchased from Taconic Biosciences. PV-Cre mice were obtained from the Jackson laboratories. Cre mice were backcrossed to C57Bl/6 mice for at least six generations. All mice were kept on a 12 hr light-dark cycle, and were group housed with ad libitum access to food and water. All animal experiments were performed according to the guidelines of the US National Institutes of Health and the Institutional Animal Care and Use Committee at the Massachusetts Institute of Technology. Experimental procedures for bouton analysis as shown in *Figure 2* were approved by the Autonoma de Madrid University ethics committee and the corresponding Madrid Regional Government agency (PROEX175/16), in accordance with the European Community Council Directive 2010/63/UE. Animals were assigned to either the MD$^{\rightarrow PL}$ or MGB$^{\rightarrow A1}$ group arbitrarily and no distinction was made with respect to gender of the animals.

## Viruses used

For retrograde monosynaptic tracing, EnvA-RVdG expressing mCherry (Titer: $1.9 \times 10^{11}$ vp/mL) was produced as previously described (*Chatterjee et al., 2018*) and generously provided by Dr. Ian Wickersham, MIT. Helper viruses AAV1-syn-FLEX-TA-TVA-GFP and AAV1-TREtight-B19G for monosynaptic tracing were also were also provided by the Wickersham lab (Titer: $1.0 \times 10^{13}$ vp/mL). Retrograde AAV expressing Cre (AAVrg-hSyn-Cre-WPRE-hGH) was sourced from Addgene vector core

(Addgene Lot#: 105553, Titer: 7.0 × $10^{12}$ vp/mL). For trans-synaptic anterograde tracing, AAV1-hSyn-Cre-WPRE-hGH (Addgene Lot# 105553, Titer: 1.0 × $10^{13}$ vp/mL), and AAV1-pCAG-FLEX-tdto-mato-WPRE (Addgene Lot# 51503, Titer: 1.0 × $10^{13}$ vp/mL) were also sourced from Penn vector core. For SSFO experiments an AAV1-CamKIIa-SSFO-GFP was sourced from UNC vector core (Titer: 1.0 × $10^{13}$ vp/mL). mGRASP labeling studies were performed using viruses (AAV2/8-CAG-pre-mGRASP-mCerulean, Titer: 2.0 × $10^{13}$ vp/mL; AAV2/8.CAG.Jx-rev.post-mGRASP-2A-dTomato, Titer: 1.0 × $10^{13}$ vp/mL) which were a kind gift from Dr. Michael Baratta, University of Colorado. For Cre expression in cortical excitatory neurons, an AAV expressing Cre under the CamKIIa promoter was sourced from Addgene vector core (pENN-AAV-CamKII-0.4. Cre-SV40, Titer: 1.0 × $10^{13}$ vp/mL).

## Surgeries for anatomical tracing studies

Mice were first anesthetized in an induction chamber receiving a continuous supply of oxygen and 5% isoflurane and then placed on a heating pad within a stereotaxic frame (Kopf Instruments, Tujunga, California). Throughout the surgery, anesthesia was maintained through continuous delivery of 1–2% isoflurane via a nose cone at a rate of 1 L/min and analgesia was provided by dual subcuta-neous injections of slow release Buprenorphine (0.5 mg/mL) and Meloxicam (5 mg/mL). The midline of the scalp was sectioned and retracted, and a small craniotomy was made over the target region. After leveling the head, a small burr hole was made over each target region using coordinates based on the mouse brain atlas of Paxinos and Franklin (2008). The coordinates are as followed (in mm from Bregma): PL: AP 2.6, ML ±0.3, DV −1.9; MD: AP −1.1, ML ±0.6, DV −3.0; A1: AP −2.92, ML ±4, DV −2.6; MGB, AP −3.0, ML ±2.05, DV −2.9 (from brain surface). For monosynaptic retro-grade-tracing experiments 300 nL of a retrograde AAVrg-hSyn-Cre-WPRE-hGH was injected into the cortex (A1 or PL) to render and 200 nL of helper AAV (1:1 mix of 5 ul AAV1-syn-FLEX-TA-TVA-GFP and AAV1-TREtight-B19G) was injected into the thalamus (MGB or MD). Three weeks later, 100 nL of RVdG-mCh(envA) was injected into the thalamus. Five days later mice were perfused, and their brains extracted for visualization. For trans-synaptic anterograde tracing of MD and MGB outputs to their respective cortices, 90 nL of AAV1-hSyn-Cre-WPRE-hGH was injected to the thalamic region of interest, and 400 nL of AAV1-pCAG-FLEX-tdtomato-WPRE was injected to the respective cortical region. Three to four weeks later, mice were perfused, and their brains were extracted for visualiza-tion. To label thalamocortical synapses onto cortical parvalbumin expressing interneurons, 75 nL of AAV-pre-mGRASP-mCerulean was injected into the MD or MGB, and 200 nL of AAV-FLEX-post-mGRASP-TdTomato was injected into the PL or A1, respectively of PV-Cre mice. To label excitatory synapses of thalamocortical innervations, 75 nL of AAV-CAG-pre-mGRASP-mCerulean was injected into the MD or MGB, and 200 nL of pENN-AAV-CamKII-0.4.-Cre-SV40 mixed (1:2) with AAV-FLEX-post-mGRASP-TdTomato was injected into the PL or A1, respectively. In both cases, mice were given 2 weeks for cells to fluoresce before perfusing animals as described in the Histology and IHC section.

Viruses were injected through a glass micropipette (Drummond Scientific) using a quintessential stereotactic injector (QSI, Stoelting, Wood Dale, Illinois). Virus was injected at a flow rate of 50 nL/min and given 10 min to spread post-injection. After the injection micropipettes were slowly retracted followed by closure of the incision. To anterogradely label thalamocortical axons from small populations of neurons located in MD or MGB, borosilicate glass micropipettes with internal glass filament (1 mm outer diameter, 4–5 μm of inner tip diameter; WPI, Sarasota, FL, USA) were backfilled with a 3% solution of lysine-fixable biotinylated dextran amine (BDA) of 10 KDa (Thermo Fisher, D1956) in PB 0.01M, pH7.4 and stereotaxically positioned over the MD or MGB. BDA was delivered by iontophoresis (positive current of 200 nA, 1 s on/off cycles) for 40 min, using a Dual Current 260 source (WPI). The micropipette was then left in place for 10 min before removal and wound closure. After a survival period of 7 days animals were sacrificed for visualization of cortical boutons.

## Histology and IHC

After viral injections were given time to express, mice were trans-cardially perfused with 30 ml of phosphate buffered saline (PBS) followed by 20 ml of 4% paraformaldehyde (PFA) in PBS. Brains were allowed to post-fix overnight at 4°C, then cryoprotected in PB 0.1M containing 30% sucrose for 24 hr. Serial 50 μm thick coronal sections were prepared using a Thermo HM550 cryotome. To

enhance the GFP signal of the TVA helper construct, immunohistochemistry was performed. Briefly, sections were first permeabilized by washing in PBS-0.3% Triton X-100. Next, sections were blocked in 10% bovine serum albumin (Sigma Millipore) in PBS-0.3% Triton X-100 for 1 hr. Then, sections were moved to 3% normal goat serum in PBS-0.1% Triton X-100 and incubated with primary chicken anti-GFP antibody (1:1000, Aves Labs, GFP1010) overnight at 4°C. After another wash, sections were incubated in the secondary Alexa Fluor 488 goat anti-chicken antibody (1:500, Thermofisher, A32931) for 2 hr at room temperature. Sections were then washed in PBS-0.1% Triton X-100 and mounted for imaging. To stain for PV expressing neurons in the cortex, brains were permeabilized and blocked as stated previously. After blocking, sections were moved to 3% bovine serum albumin in PBS-0.1% Triton X-100 and incubated with primary rabbit anti-PV (1:1000, Swant, PV-27) overnight at 4°C. After another wash, sections were incubated in the secondary Alexa Fluor 488 goat anti-Rabbit antibody (1:500, Thermo Fisher, A32731) for 2 hr at room temperature. To stain for VGluT1 and VGluT2 the same procedure was used as described for anti-PV staining using anti rabbit antibodies (VGluT1: 1:500, Synaptic Systems, 135303 and VGluT2: 1:500, Synaptic Systems, 135403). Post staining all sections were washed in PBS-0.1% Triton X-100 and mounted on glass slides and coverslipped with anti-fade mounting media (Prolong Gold, Thermo, P36930) for imaging. For all viral injections specificity of injection sites were verified using virally expressed fluorescent proteins (GFP, TdTomato, etc). For viruses where, fluorescent probes were not present in the virus itself, namely AAVrg-hSyn-Cre-WPRE-hGH and pENN-AAV-CamKII-0.4.-Cre-SV40, we co-injected a fluorescent dye which is actively taken up by cells (Hoechst 33342, Thermo Fischer, H3570), to mark the injection sites (*Figure 3—figure supplement 1*, and *Figure 5—figure supplements 1* and *2*). Animals where injection sites missed the target location were discarded and only those animals where injection sites were specific were included for further analysis and are reported as Ns for animal numbers.

To visualize BDA, sections were processed using avidin-biotine-peroxidase kit (1:100; Vectastain Elite, Vector Laboratories, Burlingame, CA, USA) and diaminodiaminobenzidine–glucose oxidase with nickel enhancement (*Veenman et al., 1992*). For a precise cytoarchitectonic delineation of thalamic nuclei and cortical layers, one series of sections was lightly counterstained with thionin, and the adjacent series was counterstained with cytochrome oxidase histochemistry (CyO) (*Wong-Riley, 1989*). Both series of sections were mounted on glass slides, dehydrated in ethanol, cleared in xylene and coverslipped with DePex (Serva, Heidelberg, Germany).

## Image analysis

All images for monosynaptic input tracing experiments were generated using a confocal microscope (LSM 710, Zeiss) with 10x/0.45 numerical aperture, 20x/0.80 numerical aperture, or 63x/1.40 numerical aperture objectives (Zeiss). Images were adjusted for contrast in ImageJ and then manually overlaid with vectorized slides from a modified version of the Reference atlas from the Allen Brain Atlas (Unified anatomical atlas) (*Chon et al., 2019*). Overlays were resized and positioned using anatomical landmarks. Input cells, within each brain area, expressing mCherry were then manually counted, using ImageJ's (NIH) cell counting plugin.

Analysis was restricted to the hemisphere where MD$^{\rightarrow PL}$ or MGB$^{\rightarrow A1}$ starter populations originated. Counts were reported as the percentage of mCherry labeled input neurons per brain area compared to the total number of mCherry+ input neurons seen across the entire brain. mCherry+ neurons seen in the MD for MD$^{\rightarrow PL}$ and in MGB for MGB$^{\rightarrow A1}$ were not included in the total counts for monosynaptic inputs. Regions that possessed more than 0.5% of the total monosynaptic inputs to MD$^{\rightarrow PL}$ or MGB$^{\rightarrow A1}$ were included in further analysis as legitimate inputs. To make quantitative comparison of the strength of input to the MD$^{\rightarrow PL}$ vs. MGB$^{\rightarrow A1}$ circuits on a single-cell basis, the total inputs within each area per animal was divided by the total number of starter cells counted in that animal and was expressed as a ratio.

To determine laminar distribution of thalamocortical innervation, all cortical sections expressing mCherry were imaged, overlaid, and counted similar to input counts. Layers were delineated using 'unified anatomical atlas' demarcations, and distance from pia was measured using ImageJ (NIH) and binned into 50 um wide bins starting from the surface of the cortex. Laminar distribution across bins were normalized to total number of cells counted in section thus analyzed across individual A1 and PL sections showing labeled neurons. Co-labeling of Parvalbumin expressing neurons with transynaptically labeled cortical neurons were found using ImageJ's *coloc* plugin (NIH), and visually validated by the experimenter.

For analysis of synapses labeled by mGRASP, PL or A1 containing sections were imaged using a confocal microscope (LSM 710, Zeiss) and 63x/1.40 numerical aperture objectives (Zeiss). Appropriate excitation wavelengths were used for different fluorescent protein markers: 458 nm for mCerulean (pre-mGRASP-labeled axons), 488 nm for GFP (mGRASP-labeled synapses), and 561 nm for TdTomato (post-mGRASP-labeled postsynaptic neurons). Multiple optical sections (0.7 µm thickness) were imaged to cover the entire z axis of the section. Postsynaptic PL or A1 neurons were imaged with appropriately adjusted spectral detectors, thus preventing potential bleed-through effects. Thereafter images were reconstructed in 3D and analyzed using Imaris Image analysis software (Imaris 9.3.2, Oxford Instruments). 3D isosurfaces (smoothness, 0.2 mm; quality level, 5) were created for each PV+ or CamKII+ neuron identified by the TdTomato signal expressed in post-mGRASP neurons and were masked to isolate the fluorescent signals within and surrounding the cell body. For each masked cell, a second round of 3D isosurfaces were created (smoothness, 0.1 mm; quality level, 7) for the mGRASP signal to create a mask around all mGRASP labeled synapses. To avoid false-positive counting of synapses, we visually confirmed that the synapses observed were made on postsynaptic cortical neurons with thalamic axons, identified with the pre mGRASP mCerulean signal, passing nearby. Care was taken to ensure that the entire mGRASP signal was covered by the isosurfaces created. The number of such isosurfaces created were used to quantify number of synapses per cell while their volumes were used to quantify the volume of the synapses.

To determine the location of BDA deposits in the thalamus and the labeled axons in the cortex, sections were systematically examined under bright-field optics using a Nikon Eclipse 600 microscope (Nikon, Tokyo, Japan). BDA-labeled axonal varicosities (putative synaptic boutons) were clearly seen in prefrontal or primary auditory cortices as ovoid axonal enlargements. To estimate the size of varicosities, maximal projection areas were measured from live images using a Nikon DMX1200 camera attached to the microscope and the NIS-Elements software tools (v3.2; Nikon) (see details in *Casas-Torremocha et al., 2019*). For each cortical area and layer, 350 varicosities were randomly selected and measured (total of 2450 varicosities). Axonal enlargements were considered as varicosities when their diameters were at least twice the adjacent axonal segment. Varicosities with maximal projection areas below the microscope resolution limit (0.2 µm2) were not included.

## Multi-electrode array construction and implantation

Custom multi-electrode array scaffolds (drive bodies) were designed using 3D CAD software (Solid-Works) and printed in Accura 55 plastic (American Precision Prototyping) as described in previous studies (*Schmitt et al., 2017*; *Wimmer et al., 2015*). Prior to implantation, each array scaffold was loaded with 16–24 independently movable micro-drives carrying 12.5 µm nichrome (California Fine Wire Company) tetrodes. Electrodes were pinned to custom-designed, 96- or 128-channel electrode interface boards (EIB, Sunstone Circuits) along with a common reference wire (A-M systems). For combined optogenetic manipulations and electrophysiological recordings, optic fibers (Doric lenses, Quebec, Canada) were embedded above or adjacent (for fibers equipped with a 45-degree mirror tip) to the electrodes. For analgesia, mice were injected with slow-release Buprenorphine (0.1 mg/kg) prior to surgery. Then animals were deeply anaesthetized with 1% isofluorane and mounted on a stereotactic frame. The animals head was shaved, and remaining hair removed with Nair. Body temperature was measured through a rectal probe and maintained using an electrical heating pad. An incision in the skin allowed access to the skull. A ~ 1.2×1.6 mm craniotomy was drilled centered at (in mm from Bregma) AP 2.0 mm, ML 0.6 mm for PFC, at AP −1.0 mm, ML 0.5 mm for MD, at AP −2.8 mm, ML 4.0 mm for A1 and at AP −3.0 mm, ML 2.0 mm, DL 3.3 mm for MGB recordings. The dura was carefully removed, and the drive implant was lowered into the craniotomy using a stereotactic arm until the shortest tetrodes touched the cortical surface. Surgilube (Savage Laboratories) was applied around electrodes to guard against fixation through dental cement. Stainless-steel screws were implanted into the skull to provide electrical and mechanical stability and the entire array was secured to the skull using dental cement. The skin was subsequently closed with Vetbond and the animal was allowed to recover on a heating blanket.

## Head fixation recordings

Simultaneous recordings from MD and PL or MGB and A1 were conducted in a custom-built setup. The head-fixation system consisted of a pair of custom 3D printed plastic fixation clamps (MakerBot Replicator, Brooklyn, NY) used to lock the implanted plastic crown at the base of the implant into place during recordings. These were fixed to an acrylic plastic frame which also supported a platform on which the animal stood. The platform was composed of low-friction acrylic and was adjusted based on the height of the animal and spring-loaded to minimize torque on the implant.

## Electrophysiological recordings

Signals from tetrodes (thalamic recordings) were acquired using a Neuralynx multiplexing digital recording system (Neuralynx, Bozeman MT) via a combination of 32- and 64-channel digital multi-plexing head stages plugged to the 32–96 channel EIB of the implant. Signals from each electrode were amplified, filtered between 0.1 Hz and 9 kHz and digitized at 30 kHz. For thalamic recordings, tetrodes were lowered from the cortex into MD −2.8 to −3.2 mm DV and into the MGB −2.8 to −3.2 mm DV. For PFC recordings, adjustments accounted for the change of depth of PL across the anterior-posterior axis. Thus, in anterior regions, unit recordings were obtained between-1.2 and −1.7 mm DV whereas for more posterior recordings electrodes were lowered −2 to −2.4 mm DV. For A1 unit recordings were obtained between −2.5 and −3.0 mm DV. Following acquisition, spike sorting was performed offline based on relative spike amplitude and energy within electrode pairs using the MClust toolbox (http://redishlab.neuroscience.umn.edu/mclust/MClust.html).

## Identification of FS and RS cells

After initial clustering, PL and A1 units were divided into fast spiking (FS) and regular spiking (RS) based on waveform characteristics and properties of the distributions of inter-spike interval (ISI) as described previously (*Halassa et al., 2014*). Features of spike waveforms such as the peak to trough time were measured for each unit across all spike waveforms. We also incorporated a measure of spike timing that has previously been used to identify FS neurons (mode of the ISI). Recorded neurons were then separated using a clustering method for three feature dimensions: 1. Half trough time 2. Peak to trough time and 3. Mode of the inter spike interval (*English et al., 2017*; *Figure 1—figure supplement 1*). Clustering across these dimensions were assessed using k-means clustering as described previously (*Nakajima et al., 2019b*).

## Firing rate analysis

For all thalamic and cortical neurons, changes in firing rate were assessed using peri-stimulus time histograms (PSTHs). PSTHs were computed using a 1 ms bin width for individual neurons in each recording session. Proportional firing rate change was calculated relative to a 500 ms long baseline where there was no optogenetic activation of the SSFO.

## Statistical analysis

Statistical analysis was performed with GraphPad Prism software (version 8.0, Prism, San Diego, California, USA). Due to the limited size of sampled data sets we assumed non-normality throughout our statistical analysis and hence performed non-parametric testing. For each statement of statistical difference included in the manuscript, a corresponding statistical comparison was performed, as mentioned in the figure legends. Briefly, we used Mann Whitney U test for all comparisons between two groups comprised of independent samples and Wilcoxon signed rank test when the samples were dependent. For comparison of cumulative distributions, the Kolmogorov-Smirnov test was used. For comparing more than two groups against each other we used the Kruskal Wallis test followed by Dunn's corrected multiple comparison. All p values are listed in the figure legends. Values are expressed as medians +/- 95% range in box-whisker plots and mean +/- SEM for bar graphs.

## Acknowledgements

We thank Ralf D Wimmer for help with electrophysiology data collection and Ian Wickersham for providing viral tools for retrograde monosynaptic tracing. We thank Laszlo Acsady for advice on mGRASP validation, Martha Bickford, Farran Briggs, Bill Guido, Judith Hirsch, Sonja Hofer, John

Huguenard, Sabine Kastner, Jeanne Paz, S Murray Sherman and W Martin Usrey for helpful feedback on study design, methods and interpretations. We also thank all members of the Halassa group for scientific discussion and support. This work was supported by funding from the European Union's Horizon 2020 Research and Innovation Programme Grant Agreement No. 945539-HBP-SGA3 and Spain's MICINN (BFU 2107–88549 P) to FC. This work was funded by NIMH grants R01MH120118 and R01MH107680 to MMH.

## Additional information

### Funding

| Funder | Grant reference number | Author |
|---|---|---|
| European Commission | 945539-HBP-SGA3 | Francisco Clasca |
| National Institute of Mental Health | R01MH120118 | Michael M Halassa |
| National Institute of Mental Health | R01MH107680 | Michael M Halassa |
| Ministerio de Ciencia e Innovación | BFU 2107–88549 P | Francisco Clasca |

The funders had no role in study design, data collection and interpretation, or the decision to submit the work for publication.

### Author contributions

Arghya Mukherjee, Conceptualization, Resources, Data curation, Formal analysis, Supervision, Validation, Visualization, Methodology, Writing - original draft, Project administration, Writing - review and editing; Navdeep Bajwa, Data curation, Formal analysis, Validation, Visualization, Methodology, Writing - original draft, Writing - review and editing; Norman H Lam, Data curation, Software, Formal analysis, Validation, Visualization, Methodology, Writing - original draft, Writing - review and editing; César Porrero, Data curation, Formal analysis, Validation, Visualization, Methodology; Francisco Clasca, Conceptualization, Resources, Supervision, Funding acquisition, Investigation, Methodology, Writing - original draft, Project administration; Michael M Halassa, Conceptualization, Resources, Supervision, Funding acquisition, Validation, Investigation, Visualization, Methodology, Writing - original draft, Project administration

### Author ORCIDs

Arghya Mukherjee  https://orcid.org/0000-0002-3341-4408
Francisco Clasca  http://orcid.org/0000-0003-0718-1337

### Ethics

Animal experimentation: All animal experiments were performed according to the guidelines of the US National Institutes of Health and the Institutional Animal Care and Use Committee at the Massachusetts Institute of Technology. Experimental procedures for bouton analysis as shown in figure 4 were approved by the Autonoma de Madrid University ethics committee and the corresponding Madrid Regional Government agency (PROEX175/16), in accordance with the European Community Council Directive 2010/63/UE.

### Decision letter and Author response

Decision letter https://doi.org/10.7554/eLife.62554.sa1
Author response https://doi.org/10.7554/eLife.62554.sa2

## Additional files

### Supplementary files
• Transparent reporting form

### Data availability
All data generated or analyzed are included in the manuscript as source data files for Figures 1 to 7.

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
