## [Decision Letter]

Thank you for submitting your article "Variation of connectivity across exemplar sensory and associative thalamocortical loops" for consideration by *eLife*. Your article has been reviewed by three peer reviewers, including Mathieu Wolff as the Reviewing Editor and Reviewer #1, and the evaluation has been overseen by Laura Colgin as the Senior Editor. The following individual involved in review of your submission has agreed to reveal their identity: John P Aggleton (Reviewer #3).

The reviewers have discussed the reviews with one another and the Reviewing Editor has drafted this decision to help you prepare a revised submission.

The work conducted here provides a comprehensive anatomical and physiological analysis of canonical higher-order (HO, with the MD as an example) and first-order (FO, with the MGB as an example) thalamic nuclei. It features an impressive toolkit indicating that the MD and the MGM have distinctive features, in large agreement with influential theoretical accounts on the differences between FO and HO nuclei. All reviewer believed that this manuscript makes an important contribution to our understanding of the heterogeneity of thalamocortical systems. They also highlighted that greater clarification on the statistical analyses conducted and on the selectivity of some of the manipulations are warranted, together with a better positioning of the data set with current state-of-the-art.

We provide below a list of essential revisions that the authors must address. Please note that the title and/or Abstract should provide a clear indication of the biological system under investigation (i.e., species name or broader taxonomic group, if appropriate). Please revise your title and/or Abstract with this advice in mind. Also please consider rearranging figures and limit the number of supplements. *eLife*'s policy as regards supplements can be found here: https://reviewer.elifesciences.org/author-guide/full (Figure supplements). By doing so, please fix Figure 5 as it is currently barely visible.

Essential revisions

1) Statistical analyses are not always comprehensive and sometimes even lacking (e.g. Figure 1—figure supplement 1, Figure 3; The “**p<0.0048” in the legend to Figure 4 is different from the significance level shown in Figure 4D which is “*”.). The Materials and methods give scant information regarding the stats used – simply saying they are “appropriate” is not sufficient. Here, we have a mixture of rank sum (Mann-Whitney presumably) and ANOVAs in the main (degrees of freedom and F values are not given, making it all the harder to unpick what was done). Please clarify all statistical analyses. There is no space issue at *eLife* so the statistical analyses need to be comprehensively described and reported in the text and all important claims must be supported by analyses or edited to be more factual (e.g. figures describing differences without analyses). In addition, with small n and multiple measures, one must be convinced that appropriate measures have been taken to avoid multiple comparisons issues.

2) Anatomical data need to be more comprehensively considered. A more thorough analysis of injection sites for tracing experiment would be appreciated, providing for instance more systematically pictures of injection sites.

Regarding the data, it is unexpected that there is so little MGB innervation of PV cells in the auditory cortex, given several studies demonstrating that MGB to PV neurons represents a dominant synaptic pathway (Schiff and Reyes, 2012, Cruikshank, Rose, Metherate, 2002). The authors should cite these references and discuss the differences between their studies and these. In addition, the authors reference Wang et al., 2013 to support thalamic innervation of PV cells, but that does not appear to be done in this article.

What is the varicosity size distribution in other layers of PL and A1 other than shown in Figure 2? From Figure 2B, L5 of PL seems to have higher axon and varicosity density than L1, but the data of it is not presented.

A further point is to carefully consider MGM subdivisions as ventral, dorsal and medial MGM portions are likely to produce different patterns of innervation of the AC. In the rabies tracing example that is given, most of the inputs come from the external cortex of the IC, suggesting that dorsal or medial division was targeted. It is also unusual that so many dorsal MGB cells are labeled in the figure, since thalamocortical cells are not known to get direct inputs from other thalamocortical neurons. It is really a critical issue since the medial division of the MGM is considered HO, not FO.

3) Theoretical account on FO and HO nuclei and novelty of the findings

The Introduction could me more straightforward as regards current conceptions on FO and HO nuclei. It should be stated more clearly that an essential difference between the two is the primary origin of the driver afferent: cortical-layer 5 for HO and the periphery for FO nuclei. By contrast all thalamic nuclei (HO and FO) are expected to receive a modulatory input from the cortex (layer 6). This is clearly summarized in multiple accounts from Guillery and Sherman that the authors are mentioning (this is very clear in Sherman, 2016 for instance). So much of the data presented are not surprising which by itself is not a problem, the data are still much welcome. But the Introduction needs to be clarified and the Discussion should more fairly reported the data based on that state-of-the-art (for example, the second paragraph of the Discussion could state that the data entirely match our expectations in term of the differential connectivity of HO versus FO nuclei.).

4) mGRASP experiment

The rationale is not entirely clear. It is stated to label the location of synapses with high resolution but when corresponding data are reported, the size of the synapses seems to be the main focus.

The mGRASP data appear to only focus on axosomatic terminals, which likely comprise a very small fraction of thalamocortical terminals, and this should be addressed. In addition, the authors examine pyramidal cells in layer 4, though it is speculated that input cells in layer 4 are spiny stellates. This part of the result section may need to be a little more comprehensive especially for the reader not aware of this technique. Expanding the text is an option.

[Editors' note: further revisions were suggested prior to acceptance, as described below.]

Thank you for resubmitting your work entitled "Variation of connectivity across exemplar sensory and associative thalamocortical loops" for further consideration by *eLife*. Your revised article has been evaluated by Laura Colgin (Senior Editor) and a Reviewing Editor.

The manuscript has been clearly improved. In particular, the new materials provided together with the more detailed statistical analyses have been much appreciated by all reviewers. However, there are some remaining issues that need to be addressed before acceptance, as outlined below:

1) The rationale for dismissing the relevance of the FO/HO perspective is still unclear. Does the present dataset provide evidence that the MD present some features of a FO nucleus?

The Discussion currently considers innervation of PV inhibitory cells (paragraph 2) and preferential innervation by cortical inputs (both layers 5 and 6 – paragraphs 3 and 4). Regarding the first point, please acknowledge the Delevich et al., 2015 study reporting similar data in dACC and discussing their relevance for a HO nucleus (the rebuttal states that this reference has been discussed but it is not the case and it is of much relevance). So all the points currently discussed are to be expected from a typical HO nucleus.

The authors should indicate in the Discussion which aspects of their data are missed by the FO/HO perspective or acknowledge that the present results are in line with this perspective.

2) The authors should carefully proofread the paper. As stated in the previous decision letter, the title needs to mention the biological system used. A number of statements need to be supported by references from the literature (e.g. that rabies suffers from false-positives and false negatives, that AAV1 may have bidirectional transport).

---

## [Author Response]

Essential revisions1) Statistical analyses are not always comprehensive and sometimes even lacking (e.g. Figure 1—figure supplement 1, Figure 3; The “**p<0.0048” in the legend to Figure 4 is different from the significance level shown in Figure 4D which is “*”.). The Materials and methods gives scant information regarding the stats used – simply saying they are “appropriate” is not sufficient. Here, we have a mixture of rank sum (Mann-Whitney presumably) and ANOVAs in the main (degrees of freedom and F values are not given, making it all the harder to unpick what was done). Please clarify all statistical analyses. There is no space issue at eLife so the statistical analyses need to be comprehensively described and reported in the text and all important claims must be supported by analyses or edited to be more factual (e.g. figures describing differences without analyses). In addition, with small n and multiple measures, one must be convinced that appropriate measures have been taken to avoid multiple comparisons issues.

We thank the reviewers for pointing out the inconsistencies with statistical reporting in the earlier version of the manuscript, to directly address their feedback, we have now:

1) Corrected the typographical errors and included the missing statistics for Figure 1—figure supplement 1, Figure 3E. Specifically, it was an error on our part to use terms such as rank-sum and one way ANOVA on ranks, rather than the more specific and proper terms which we have now substituted as follows: Mann Whitney U test and Wilcoxon signed rank test (instead of rank sum test, respectively, for unpaired and paired comparisons), Kruskal-Wallis test (instead of one way ANOVA on ranks) and Kolmogorov-Smirnov test to compare frequency distribution plots.

2) Clarified in the Materials and methods that we have not assumed that any of our datasets are normally distributed and therefore have used nonparametric statistical tests for this study. Thus, we don’t report degrees of freedom or F values as usual for parametric tests such as ANOVA. To avoid multiple comparison issues, we employed post hoc Dunn’s correction for further analysis. We have also added these details about the statistical tests in Materials and methods section of the manuscript.

2) Anatomical data need to be more comprehensively considered. A more thorough analysis of injection sites for tracing experiment would be appreciated, providing for instance more systematically pictures of injection sites.

We thank the reviewers for bringing up this important point, which we now directly address as:

1) For all experiments virus injection sites were carefully evaluated with expression of a fluorescent protein (eg: EYFP, GFP, TdTomato etc) and only those where the fluorescent protein expression was restricted to the area of interest was considered for further analysis (as also mentioned in Histology and IHC subsection under Materials and methods).

2) For virus injections where, fluorescent probes were not present in the virus itself, namely AAVrghSyn-Cre-WPRE-hGH and pENN-AAV-CamKII-0.4.-Cre-SV40, we co-injected a fluorescent dye which is actively taken up by cells (Hoechst 33342), to mark the injection sites (See Figure 3—figure supplement 1, and Figure 5—figure supplements 1, 2).

3) We have now included schematics depicting the injection site for each animal that contributed to the data set for that experiment and carefully refer to injection site images and schematics throughout the Results section of the manuscript.

Regarding the data, it is unexpected that there is so little MGB innervation of PV cells in the auditory cortex, given several studies demonstrating that MGB to PV neurons represents a dominant synaptic pathway (Schiff and Reyes, 2012, Cruikshank, Rose, Metherate, 2002). The authors should cite these references and discuss the differences between their studies and these. In addition, the authors reference Wang et al., 2013 to support thalamic innervation of PV cells, but that does not appear to be done in this article.

Thank you for these references and raising this important point. We apologize for giving the impression that our data suggest that the MGB has little A1 PV innervation. To address this important issue, we now:

1) Clarify that our data shows that the MD innervates PL PV neurons with a larger frequency than MGB innervation of A1 PV neurons (l 185) as follows:

“Our results are in line with previous results indicating substantial inhibitory innervation of A1 PV neurons by MGB afferents, (de La Rocha et al., 2008), but additionally indicate that the proportional innervation of these inhibitory neurons in the PL by thalamic afferents is even larger.”

2) Clarify that the point above, in conjunction with the data showing MGB neurons target layer 4 excitatory neurons with large synapses (consistent with previous findings (Llano and Sherman, 2008)), led us to hypothesize (l311-315) that the functional effect of activating MD vs MGB is distinct in their respective cortical targets, perhaps with respect to E/I balance.

3) We have also replaced the erroneous Wang et al., 2013 reference with that from de La Rocha et al., 2008 to support thalamic innervation of FS neurons.

4) Lastly, we now clarify in the Discussions that MD innervation of PL is not exclusively inhibitory (as one may conclude based on some of our framing). Again, this is another nuanced point that we emphasize:

“However, we should also emphasize that this is not the only functional effect observed. Previous studies, including our own, indicate that associative thalamic drive enhances lateral effective connectivity in cortex required for maintaining working memory and attentional control. This effect is still observed in our dataset here (data not shown).”

What is the varicosity size distribution in other layers of PL and A1 other than shown in Figure 2? From Figure 2B, L5 of PL seems to have higher axon and varicosity density than L1, but the data of it is not presented.

We thank the reviewers for bringing up this important point. We now present the layer-wise differences in PL and A1 of axon and varicosity densities of thalamocortical projections originating from MD and MGB respectively. These results show that the MD boutons in all layers of the PL and MGB boutons in Layers 1, 3 5 of A1 are consistently smaller in size but larger in frequency than MGB boutons in layer 4 of A1. However, we do not see L5 of PL having higher thalamocortical varicosity density than L1 (Figure 2H).

A further point is to carefully consider MGM subdivisions as ventral, dorsal and medial MGM portions are likely to produce different patterns of innervation of the AC. In the rabies tracing example that is given, most of the inputs come from the external cortex of the IC, suggesting that dorsal or medial division was targeted. It is also unusual that so many dorsal MGB cells are labeled in the figure, since thalamocortical cells are not known to get direct inputs from other thalamocortical neurons. It is really a critical issue since the medial division of the MGM is considered HO, not FO.

Thank you for raising this critical issue. To directly address it we:

1) Clarify that our anterograde tracing, which is the most critical set of anatomical studies supplementing the functional ones were mostly restricted to the MGB ventral division, a classical FO nucleus. We now show these results in the following new Supplemental Figures (Figure 3—figure supplement 1).

2) We highlight the limitations of our retrograde rabies tracing which was hard to localize to the MGB ventral division and thereby those data may contain a HO component as the reviewers pointed out (l349353). In fact, we had already shown this in our previous manuscript (Figure 5—figure supplement 2), where the starter population is only biased towards the FO ventral MGB (~65% of all starter neurons), with some leak into the HO (~20%) medial and dorsal MGB. However, that said, the example of MGB inputs originating from the external shell of the IC were an oversight, and not representative of the dominant pathway from the IC. We now replace this image with the more dominant central IC input. Again, we thank the reviewers for pointing out this oversight.

3) Theoretical account on FO and HO nuclei and novelty of the findingsThe Introduction could me more straightforward as regards current conceptions on FO and HO nuclei. It should be stated more clearly that an essential difference between the two is the primary origin of the driver afferent: cortical-layer 5 for HO and the periphery for FO nuclei. By contrast all thalamic nuclei (HO and FO) are expected to receive a modulatory input from the cortex (layer 6). This is clearly summarized in multiple accounts from Guillery and Sherman that the authors are mentioning (this is very clear in Sherman, 2016 for instance). So much of the data presented are not surprising which by itself is not a problem, the data are still much welcome. But the Introduction needs to be clarified and the Discussion should more fairly reported the data based on that state-of-the-art (for example, the second paragraph of the Discussion could state that the data entirely match our expectations in term of the differential connectivity of HO versus FO nuclei.).

We thank the reviewer for highlighting this point, but we would like to ask that we keep our formulation with some clarification. The reason is that we are trying to emphasize that the thalamus may be better described as individual motifs that maybe distributed across different thalamic nuclei in different proportions. For example, the classical notions of HO and FO, or core and matrix may apply more to individual motifs that nuclei as a whole. In that manner, the MD may have both HO and FO characters for example, as can be observed in classical (Goldman-Rakic and Porrino, 1985), and contemporary work (Peng et al., n.d.).

4) mGRASP experimentThe rationale is not entirely clear. It is stated to label the location of synapses with high resolution but when corresponding data are reported, the size of the synapses seems to be the main focus.

Thank you for raising this. We have now removed this erroneous claim.

The mGRASP data appear to only focus on axosomatic terminals, which likely comprise a very small fraction of thalamocortical terminals, and this should be addressed.

We are aware of this excellent issue raised by the reviewer. This caveat is now elaborated in the Discussion section of the manuscript as a limitation.

In addition, the authors examine pyramidal cells in layer 4, though it is speculated that input cells in layer 4 are spiny stellates. This part of the result section may need to be a little more comprehensive especially for the reader not aware of this technique. Expanding the text is an option.

Thank you for raising another excellent point. We now highlight the species differences with respect to finding spiny stellate cells in layer 4 cortex. Specifically, in rodents and outside of primary somatosensory cortex, spiny stellate cells are rare (Barbour and Callaway, 2008; Sakata and Harris, 2009; Smith and Populin, 2001), and therefore pyramidal cells are the dominant excitatory cell type in mouse A1 (l214-215).

[Editors' note: further revisions were suggested prior to acceptance, as described below.]

The manuscript has been clearly improved. In particular, the new materials provided together with the more detailed statistical analyses have been much appreciated by all reviewers. However, there are some remaining issues that need to be addressed before acceptance, as outlined below:1) The rationale for dismissing the relevance of the FO/HO perspective is still unclear. Does the present dataset provide evidence that the MD present some features of a FO nucleus?

We appreciate that the reviewers' have asked for a clearer clarification on the relationship between our data and the notion of first vs. higher order thalamic nuclei. We also apologize for giving the impression that we had been dismissive of this comment earlier and now clarify in the text what, in our minds, the relationship is by adding the following:

1) Addition to the text (Introduction):

“Based on the origin of their driving inputs our data is consistent with the classification of the MD as a higher order nucleus and the MGB as a first order nucleus (Mitchell, 2015; Sherman and Guillery, 1998).”

2) Addition to the text (new section in the Discussion):

“The original formulation of first vs. higher order thalamic nuclei is primarily related to how neurons within a thalamic nucleus are driven (i.e. whether their inputs are from sensors or cortex; (Sherman and Guillery, 1998)). (…) However, a classification scheme that incorporates both inputs and outputs of single thalamic neurons would be most helpful in constraining functional models in the future (Halassa and Sherman, 2019).”

The Discussion currently considers innervation of PV inhibitory cells (paragraph 2) and preferential innervation by cortical inputs (both layers 5 and 6 – paragraphs 3 and 4). Regarding the first point, please acknowledge the Delevich et al., 2015 study reporting similar data in dACC and discussing their relevance for a HO nucleus (the rebuttal states that this reference has been discussed but it is not the case and it is of much relevance). So all the points currently discussed are to be expected from a typical HO nucleus.

We have added the Delevich et al., 2015 study as follows: “Notably and in line with a previous study (Delevich et al., 2015), the MD⃗^PL^ neurons also make more frequent contacts with local inhibitory parvalbumin+ (PV+) neurons in the prefrontal cortex compared to MGB⃗^A1^ neurons in the A1.”

The authors should indicate in the Discussion which aspects of their data are missed by the FO/HO perspective or acknowledge that the present results are in line with this perspective.

We have acknowledged that none of our data are inconsistent with the FO/HO perspective. However, as pointed out above, the FO/HO perspective does not capture the output component of our data (since, by definition, it is an input-based classification). Similarly, we had also avoided discussing output-based classifications such as core vs. matrix, which we now also mention in the Discussion. We appreciate that this point was raised further and think that addressing it does provide the manuscript an added perspective.

2) The authors should carefully proofread the paper. As stated in the previous decision letter, the title needs to mention the biological system used. A number of statements need to be supported by references from the literature (e.g. that rabies suffers from false-positives and false negatives, that AAV1 may have bidirectional transport).

We have corrected the errors above and have proofread the document.